# $\alpha$-PFN: Fast Entropy Search via In-Context Learning

## Abstract

Information-theoretic acquisition functions such as Entropy Search (ES) offer a principled exploration–exploitation framework for Bayesian optimization (BO). However, their practical implementation relies on complicated and slow approximations, i.e., a Monte Carlo estimation of the information gain. This complexity can introduce numerical errors and requires specialized, hand-crafted implementations. We propose a two-stage amortization strategy that learns to approximate entropy search-based acquisition functions using Prior-data Fitted Networks (PFNs) in a single forward pass. A first PFN is trained to be conditioned on information about the optima; second, the $\alpha$-PFN is trained to predict the expected information gain by training on information gains measured with the first PFN. The $\alpha$-PFN offers a scalable and learnable approximation, which replaces the complex approximations with a single forward pass per candidate, enabling rapid and extensible acquisition evaluation. Empirically, our approach is competitive with state-of-the-art entropy search implementations on synthetic and real-world benchmarks while accelerating the different entropy search variants by over at least a factor of 12x, with the largest speed ups around 30x for the highest 8 dimensional problems.

## 1 Introduction

Bayesian Optimization (BO) is a method to maximize the output of a black-box function with as few as possible consecutive trials. It is especially beneficial when function evaluations are costly. BO finds use in various fields (Shahriari et al., 2015), such as optimizing the hyperparameters of large neural networks (Snoek et al., 2012; Feurer and Hutter, 2019). Here, the black-box function is the performance on a validation set given hyperparameter specifications, which necessitates a full training run for evaluation. It is thus of utmost importance to maximize performance in as few trials as possible. To achieve this, the canonical BO framework maintains a probabilistic regression surrogate model of the black-box function fit to the performance observations thus far, and maximizes an acquisition function quantifying the exploration-exploitation trade-off for the given posterior.

The information-theoretic acquisition functions (Villemonteix et al., 2009), such as Entropy Search (ES, Hennig and Schuler, 2012), offer a principled way to perform global optimization of Gaussian Process (GP) surrogate models (Williams and Rasmussen, 2006) by selecting queries that maximize the expected information gain regarding the location of the optimum. Although ES offers an elegant framework, its performance and runtime are hampered by complex handcrafted approximation schemes, as there is no simple analytical definition for GPs, unlike for classical acquisition functions e.g., Expected Improvement (EI; Močkus, 1975).

Many efforts have been made to improve Entropy Search's computational efficiency and performance with GPs. Hernández-Lobato et al. (2014) show that it is possible to rewrite the Entropy Search acquisition function to depend on terms that are more tractable to estimate and approximate, resulting in an acquisition function which they call Predictive Entropy Search (PES). Wang and Jegelka (2017) proposed Max-value Entropy Search (MES) and most recently Hvarfner et al. (2022) and Tu et al. (2022) proposed Joint Entropy Search (JES). These methods adjust the original Entropy Search formulation to enable more efficient approximations and/or better BO performance. All of these improvements to ES still rely on handcrafted, sampling-based approximations, though. The runtime of BO is becoming an increasingly important issue, as practitioners also apply BO to

blackbox functions that are faster to query (so-called high-throughput optimization; Eriksson et al., 2019; Daulton et al., 2022; Maus et al., 2022).

Recently, it has been shown that Prior-Data Fitted Networks (PFN; Müller et al., 2021), a transformer-based conditional neural process (Garnelo et al., 2018) can be used to speed up training and prediction of Gaussian Process regression models (Müller et al., 2023). The key idea is to use transformers to approximate posteriors for a prior: the transformer accepts the training data and test inputs, and outputs the posterior predictive distribution for each test sample. By training on millions of datasets sampled from a prior, the transformer meta-learns to perform Bayesian prediction in a single forward pass. Inference can be computationally much more efficient compared to fully-Bayesian GPs, which require the use of Markov Chain Monte Carlo (MCMC; Robert et al., 1999). Note that training PFNs has one-time upfront costs (on the order of a GPU day) — but these are amortized over millions of applications to different optimization problems, as inference is fast.

We explore how the same technique can be applied to approximate quantities used in Entropy Search. The Entropy Search acquisition functions (PES, MES and JES) all require many Monte Carlo samples and other manual approximations. Instead of deriving another approximation, our transformer learns to approximate the acquisition functions in a single forward pass. To avoid these costly samples, *we replace sampling by learning* by training a second transformer, $\alpha$-PFN, which directly predicts the acquisition, amortizing the cost of these MC samples. We evaluate the $\alpha$-PFN's performance against existing sampling-based approximations of PES, MES, and JES in terms of optimization quality (inference regret) and computational cost (optimization runtime). Our experiments cover two settings: Bayesian optimization with fully Bayesian GPs on synthetic benchmark functions and on real-world black-box hyperparameter optimization tasks from LC-Bench.

## 2 BACKGROUND, NOTATION AND RELATED WORK

In this section, we introduce key concepts and related work.

### 2.1 BLACK-BOX OPTIMIZATION AND BAYESIAN OPTIMIZATION (BO)

In Black-box Optimization (BBO), the goal is to maximize a function $f(x)$ over a domain $A$. We denote $x^* = \arg\max_{x \in A} f(x)$ and $f^* = f(x^*)$. We assume for simplicity that $A = [0, 1]^d$. We only have black box access to $f(x)$, i.e., we can only obtain function evaluations which are typically corrupted with noise: $y = f(x) + \epsilon$. BBO is iterative: in each iteration $t$ a query $x_t$ is made to the black box function and the corresponding observation $y_t$ is revealed. In Bayesian Optimization (BO), a Bayesian surrogate model, such as Gaussian Process or transformer, is fitted to the previously collected data, i.e., it is fitted on the training data $D_{trn} = \{(x_1, y_1), \ldots, (x_t, y_t)\}$ to predict the Posterior Predictive Distribution (PPD) $p(y_{tst}|D_{trn}, x_{tst})$ for a test point $x_{tst}$. BO uses an acquisition function that operates on the predictive distribution and decide which point to query next. We write it as $\alpha = \alpha(x, D_{trn})$, the next query is then determined as $x_{t+1} = \arg\max_{x \in A} \alpha(x, D_{trn})$.

### 2.2 ENTROPY SEARCH (ES)

The original Entropy Search (ES) method selects queries by maximizing the expected reduction in the entropy of the optimum location $x^*$ (Hennig and Schuler, 2012):

$$\alpha_{ES}(x, D_{trn}) = H(p(x^*|D_{trn})) - \mathbb{E}_{y \sim p(y|D_{trn}, x)}\left[H(p(x^*|D_{trn} \cup \{(x, y)\}))\right]. \quad (1)$$

Here, $H(p(x^*|D_{trn}))$ represents the entropy of the posterior distribution over the optimum location given the observed data $D_{trn}$. The first term is constant with respect to query selection and can be ignored during optimization. The second term represents the expected entropy after obtaining the new observation $(x, y)$, where the expectation is taken over the predictive distribution $p(y|D_{trn}, x)$.

Computing $\alpha_{ES}$ exactly is generally intractable, as the entropy of the GP's maximizer does not have a closed-form expression and requires averaging over multiple samples of $y$. To approximate this, Hennig and Schuler (2012) propose two methods: Monte Carlo sampling and Expectation Propagation (Minka, 2001). However, both approaches are computationally expensive, making ES impractical for many real-world applications.

**Predictive Entropy Search (PES).** The entropy reduction in ES can be expressed in terms of mutual information (MI) between $x^*$ and $y$ conditioned on $D_{trn}$. Utilizing the symmetry of the MI in $x^*$ and $y$, Hernández-Lobato et al. (2014) propose to optimize

$$\alpha_{PES}(x, D_{trn}) = H(p(y|D_{trn}, x)) - \mathbb{E}_{x^* \sim p(x^*|D_{trn})}\left[H(p(y|D_{trn}, x, x^*))\right] \tag{2}$$

which results in the objective as ES (Equation 1) but allows more efficient approximation. The first term is analytically tractable since the posterior of the GP is Gaussian. Hernández-Lobato et al. (2014) propose a sequence of approximations to accurately estimate the entropy in the second term, and the outer expectation over $x^*$ is distributed according to $p(x^*|D_{trn})$. To obtain these, sample paths from the GP posterior are approximated using random Fourier features (RFF, Rahimi and Recht, 2007). Each sample path is maximized to obtain draws of the posterior over $x^*$. The predictive posterior $p(y|D_{trn}, x^*)$ cannot be obtained exactly. Thus, conditioning on tractable alternatives, such as convexity at $x^*$, and constraints such as $f(x^*) \geq \max_{i \in [1,t]} y_i$, serves to approximate it.

**Max-value Entropy Search (MES).** Max-value Entropy Search (MES; Wang and Jegelka, 2017) aims to reduce uncertainty over the maximum function value, $f^*$, by selecting the query point that maximizes the expected information gain:

$$\alpha_{MES}(x, D_{trn}) = H(p(y|D_{trn}, x)) - \mathbb{E}_{f^* \sim p(f^*|D_{trn})}, \left[H(p(y|D_{trn}, x, f^*))\right] \tag{3}$$

Here, the expectation is taken over the posterior distribution of $f^*$ given $D_{trn}$. In addition to the RFF sampling approach for $f^*$ and $x^*$ proposed by Hernández-Lobato et al. (2014), Wang and Jegelka (2017) propose a simpler alternative for $f^*$ using a Gumbel distribution. Compared to PES, MES reduces the expectation from $d$ dimensions to one. Moreover, they assume that $p(y|D_{trn}, x, f^*)$ can be well-approximated by a truncated normal distribution, which enables an analytical entropy calculation. However, this assumption holds only in noiseless settings (Takeno et al., 2020; Nguyen et al., 2022). Due to the simpler approximation, MES is substantially faster than PES, and has seen multiple extensions, e.g. to parallel queries (Moss et al., 2021) (GIBBON-MES), noisy (Takeno et al., 2020; 2022) and multi-fidelity (Moss et al., 2021) problems.

**Joint Entropy Search (JES).** Either the distribution $p(x^*|D_{trn})$ or $p(f^*|D_{trn})$ might only provide a limited view of the posterior uncertainty over the optimum. Therefore Hvarfner et al. (2022) and Tu et al. (2022) propose to reduce the uncertainty on the joint distribution of the maximum value and its location:

$$\alpha_{JES}(x, D_{trn}) = H(p(y|D_{trn}, x)) - \mathbb{E}_{(x^*, f^*) \sim p(x^*, f^*|D_{trn})}\left[H(p(y|D_{trn}, x, x^*, f^*))\right] \tag{4}$$

The expectation is approximated by sampling in the same manner as PES. For each sample, the pair $(x^*, f^*)$ is added to the GP's training set so that the posterior can be updated using regular GP machinery, conditioning either on a *noiseless* optimal value $f^*$ (Hvarfner et al., 2022), or a $y^* = f^* + \varepsilon$ containing observation noise (Tu et al., 2022). Both Hvarfner et al. (2022) and Tu et al. (2022) use a local constraint to condition on the maximum similar to MES. The resulting extended skew distribution (Nguyen et al., 2022; Hvarfner et al., 2022; 2023) does not admit a closed form for the entropy, and is therefore approximated either by Monte Carlo sampling of the integral (Tu et al., 2022) or moment matching with a Gaussian (Moss et al., 2021; Hvarfner et al., 2022), lower bounding the MI (Moss et al., 2021).

**Fully Bayesian treatment for PES, MES and JES.** In practice, one may not know the ideal hyperparameters for the GP. A principled Bayesian would set hyperpriors on the hyperparameters, and integrate them out whenever possible. Because these integrals are expensive, typically approximations are used. Hernández-Lobato et al. (2014) were the first to give a fully Bayesian approximation to the acquisition function for ES (and Snoek et al. (2012) gave the first treatment in the BO literature). Hernández-Lobato et al. (2014) use slice sampling (Vanhatalo et al., 2013) to sample from the posterior over the GP hyperparameters, and this has been applied also for JES and PES (Hvarfner et al., 2022). The acquisition function is computed for each of these hyperparameter samples from the posterior and averaged. It would be more Bayesian to integrate out the hyperparameters first, and then compute a single acquisition function for the fully Bayesian model. However, such a fully Bayesian model would consists of superposition of Gaussians, making this approach computationally hard. Our $\alpha$-PFN makes this approach computationally tractable.

## 2.3 PRIOR-DATA FITTED NETWORKS (PFNs)

Prior-data Fitted Networks (PFNs; Müller et al., 2021) are transformer neural networks that learn to perform Bayesian predictions in a single forward pass by training on synthetic data sampled from a predefined prior $p(D)$. The prior defines a distribution over datasets $D$ of input and output pairs $(x, y)$. During training, datasets from the prior are split into two parts: a training and test set. The transformer takes the training set pairs $(x, y)$ as input (indicated as $D_t rn$), and is trained to predict the correct outputs for the test set inputs. For simplicity of notation we always assume a test set of size one, and indicate it by $x_{tst}$ and $y_{tst}$. The transformer outputs a distribution for each test object, and is trained with cross entropy. It can be shown that the transformer will approximate the posterior predictive distribution $p(y_{tst}|x_{tst}, D_{trn})$ for any prior $p(D)$ when trained on prior samples (Müller et al., 2021). At inference time, the training set and (unlabeled) test set are provided as input to the transformer, which then predicts test targets. Note that no gradient-descent takes place at inference time, but the PFN learns from new data in-context (during the forward pass).

The versatility of this paradigm, has led to a variety of applications, such as Bayesian Optimization (Müller et al., 2023; Rakotoarison et al., 2024), forecasting time-series (Dooley et al., 2024) and learning curves (Adriaensen et al., 2024; Viering et al., 2024); and as a foundation model for tabular data (Hollmann et al., 2025). Most relevant to our work, PFNs have been used to accurately approximate Gaussian Process regression (Müller et al., 2021; 2023).

In this work, we use the TabPFNv2 architecture (Hollmann et al., 2025), which improves upon v1 of Müller et al. (2023). Specifically, v1 requires zero-padding if features are not used, and can only be applied to dimensionalities seen during training. The v2 architecture processes each scalar cell of tabular data individually (a cell refers to a scalar value $x_i$ or $y$), encoding each to one embedding vector. Note that encoders for each $x_i$ are the same, but the encoder for $y$ is different. The laters of this architecture contain of three components: i) a self-attention across features and y within each sample, ii) a self-attention between samples for each feature separately, and iii) a per-embedding MLP that processes each embedding seperately.

## 2.4 TRANSFORMERS FOR BLACK-BOX OPTIMIZATION

Multiple previous works have proposed to use transformers for black-box optimization. Chen et al. (2022) proposed the use of transformers to predict points to be selected for Bayesian optimization, based on empirical traces of other Bayesian optimization algorithms utilizing handcrafted acquisition functions such as EI. Chen et al. (2017) also use GP sample paths and meta-learn acquisition functions, but do not focus on information-theoretic functions. Tiao et al. (2021) and later works by Song et al. (2022) propose the use of binary classification models to directly predict acquisition functions such as probability of improvement or expected improvement. Following a similar idea, end-to-end approaches Volpp et al. (2020); Maraval et al. (2023) have also been proposed to simultaneously learn both the acquisition function and the surrogate model from scratch using reinforcement learning from trajectories. However, these latter two works focus on the transfer learning BO setting, where it is assumed that a family of related functions are available to learn from. In contrast, we do not use any transfer learning.

Concurrently with this work, Chang et al. (2024) propose an architecture to perform inference conditional on latent variables, such as the maximum of a GP. They use a similar architecture as (Müller et al., 2021) to condition on $f^*$ and predict $f^*$ and then use monte-carlo sampling at inference time to compute MES. In this work, we not only remove the slow monte-carlo sampling at inference time by amortizing the acquisition computation itself, but also extend the approach to PES and JES. This is naturally harder for their approach, because they would need to predict $x^*$, which is higher-dimensional and requires autoregressive decoding due to strong dependencies.

Recently, Hu et al. (2024) proposed a transformer called InfoNet for amortizing computations regarding mutual information. Our work is similar, since acquisition functions for entropy search are also defined in terms of mutual information. However, the second stage amortizing using our Alpha-PFN offers even more speed-up compared to applying InfoNet for computing acquisitions, as we don't need to approximate the expectation over optima. Surrogate models can also find applications in bandits, see for example (Zhou et al., 2020).

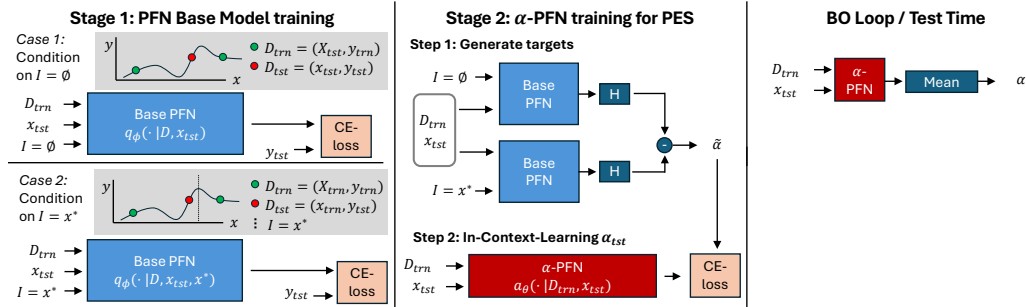

Figure 1: An overview of our pipeline. Left: we illustrate 2 out of 4 cases for the base PFN training. Middle: how to train the $\alpha$-PFN for PES. Right: how to use the $\alpha$-PFN at test-time.

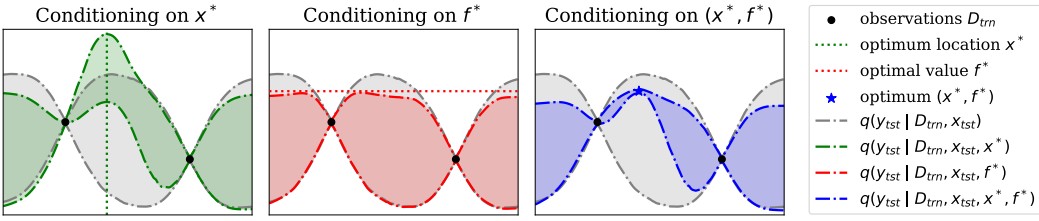

Figure 2: Our base PFN estimates the Posterior Predictive Distribution (PPD) conditioned on different types of information regarding the optimum: none (unconditional case), $x^*$, $f^*$, or both.

## 3 ENTROPY SEARCH WITH PFNS

In the following we explain how we train $\alpha$-PFN, which predicts the acquisition values for PES, MES, and JES directly. See Figure 1 for an overview of the training setup. To generate training data for the $\alpha$-PFN, we additionally train a traditional (base) PFN. The base PFN is trained to make predictions conditioned on information of the location and/or value of the optimum. This base PFN can compute $p(y|D_{trn}, x, I)$ ($I = x^*$ for PES, $I = f^*$ for MES, and $I = (x^*, f^*)$ for JES). We use its predictions to compute a sample from the estimates in the definitions of the acquisition functions by conditioning it on information on the datasets optimum. We now train the second PFN model, the $\alpha$-PFN, which directly predicts the acquisition function, using the information gain computed with the base PFN. Crucially, we condition the base model on precomputed $x^*$ and/or $f^*$, avoiding any MC-samples, but the $\alpha$-PFN is only conditioned on data available at inference time. Training $\alpha$-PFN therefore amortizes the cost of approximating the expectation by Monte Carlo samples.

**Pre-computing Gaussian Process prior data.** To construct our PFN models for GP inference we need to train the model on millions of dataset samples from a GP prior. Furthermore, we need to know $x^*$ and $f^*$ for each dataset, which is not feasible to compute for an exact GP sample. To make this feasible, we approximate the GP samples using Random Fourier Features (RFFs; Rahimi and Recht, 2007). The results of this precomputation are (approximate) samples, that can be queried efficiently for arbitrary $x$, on which we can employ gradient-based optimization to find approximate $x^*$ and $f^*$. See Appendix B for more details.

**Training the base PFN to condition on information on the optimum.** When we are training on our precomputed RFF GP data, we have access to $x^*$ and $f^*$, and feed these to the PFN to condition on them. We add one extra data point to the context of the PFN, which is encoded just like the other data points but using a different encoder, so that the PFN will learn to treat it differently. We randomly pass $x^*$ and $f^*$ with a $50\%$ chance such that a single model is able to handle all four cases, and if one is not passed, we instead pass a learned embedding to indicate it is missing. This

way we train our base PFN $q(y|x, D_{trn}, I)$, where $I$ can contain $x^*$ and/or $f^*$. For more details see Appendix C.

**Training $\alpha$-PFN.**  After training the base model, we train a second PFN model, the $\alpha$-PFN, that takes the observation data $D$ and query point $x$ as input and predicts the acquisition $\alpha(x, D)$ directly. We train $\alpha$-PFN to use $D_{trn}$ and $x$ to predict

$$H(q(y|D_{trn}, x)) - H(q(y|D_{trn}, x, I)), \tag{5}$$

where $I$ are extra information on the maximum. The (differential) entropy is computed analytically on the PFNs Riemann distribution (Müller et al., 2021). If we are training a PES-PFN, we have $I = x^*$, and $I = f^*$ for MES-PFN and $I = (x^*, f^*)$ for JES-PFN. [1] During training, we can just feed in the $x^*$ and/or $f^*$ that were precomputed for our GP data. This info is only used to determine the prediction target during training and is *not required at test time for the $\alpha$-PFN*. $\alpha$-PFN will learn the distribution of this information gain, as this is a random variable, where the varying factor is the location and/or value of the optimum. The mean of this distribution that $\alpha$-PFN aims to learn coincides with the PES/MES/JES acquisition in equations 2/3/4. We will formally prove this in the next section. This way, the $\alpha$-PFN avoids the need to sample $x^*$ and $f^*$ at test-time via MC-samples. For more training details see Appendix D.

**Handling domain shift at optimization time.**  Müller et al. (2023) train PFNs for BO by sampling inputs $x$ uniformly from the domain. However, during the process of BO, the query points $x$ tend to cluster around local optima. This domain shift in the pretraining procedure appears to adversely affect BO performance of PFNs in higher dimensions. To combat this, we train both base and $\alpha$ PFN with a sampling procedure that mimics the clustering behavior observed during the BO process. See Appendix F for more detail how we implemented a fast heuristic to generate approximate BO-traces.

**Fully-Bayesian models.**  To approximate ES for a fully-Bayesian GP, a GP with a priors over its hyperparameters, we simply sample its hyperparameters before sampling from the GP at train time. This is the only minor change needed in our training pipeline and incurs virtually no additional overhead. When trained on fully-Bayesian data, our base model $q$ directly integrates out the uncertainty over the GP's hyperparameters. In contrast to the classical GP approach, we only compute one acquisition function, derived from information gain of this fully Bayesian base model. This acquisition allows our ES-PFN to sample points to reduce the uncertainty with respect to the hyperparameters as well. As such, we can expect the fully-Bayesian $\alpha$-PFN to query points more efficiently in the fully-Bayesian case than the GP counterparts, since the GP ES variants ignore uncertainty over hyperparameters when computing the acquisition function.

## 4   $\alpha$-PFN Approximates Entropy Search

We derive in what way $\alpha$-PFN approximates entropy search. To train $\alpha$-PFN, we use an additional PFN, our base model $q(y|x, D, I)$, where $D$ is the observation data, $x$ is the currently considered point and $I$ is extra information about the maximum of the function underlying $D$. We train the base model with a standard PFN objective, but additionally feed $I$ to it. The proof that $q$ approximates $p(y|x, D, I)$ is very similar to the PFN proof by Müller et al. (2021) and is given in Appendix A.

The actual $\alpha$-PFN $a_\theta(\cdot|D, x)$ takes $D$ and $x$ as input, and outputs a distribution, which has the different ES acquisition functions as its mean (depending on $I$). $\alpha$-PFN is trained to minimize

$$l_\theta = \mathbb{E}_{D,x,I} \left[ -\log a_\theta(\tilde{\alpha}(D, x, I)|x, D) \right], \tag{6}$$

where $\tilde{\alpha}(D, x, I) = H(q(y|D, X)) - H(q(y|D, x, I))$. To understand why this is the right loss, we re-examine the entropy search acquisition functions. See Section 2.2 for further reference on the definition of entropy search. Assuming $q$ approximates $p$ exactly, we can write PES, MES, JES as

$$\mathbb{E}_{\tilde{\alpha}\sim p(\tilde{\alpha}|D,x)}[\tilde{\alpha}] = \mathbb{E}_{I\sim p(I|D,x)}[\tilde{\alpha}(D, x, I)] \tag{7}$$

$$= \mathbb{E}_{I\sim p(I|D,x)}[H(q(y|D, X)) - H(q(y|D, x, I))], \tag{8}$$

---

[1]Note that, we predict the information gain of $y$, the noisy target, consistent with the original PES formulation (Hennig and Schuler, 2012; Hernández-Lobato et al., 2014).

where $I = x^*$ for PES, $I = f^*$ for MES and $I = (x^*, f^*)$ for JES. The change of variable in Equation 7 is justified by the law of the unconscious statistician, as we are not considering the density of $\tilde{\alpha}$ outside an estimate.

**Insight 1.** *The objective $l_\theta$ is equal to the KL divergence between $p(\tilde{\alpha}|D, x)$ and the $\alpha$-PFN's output up to a constant. We can see that the mean of $\alpha$-PFN's output distribution can then be used to obtain entropy search approximations, like in the acquisition function definition in Equation 7.*

*Proof.* This can be shown as follows

$$l_\theta = \mathbb{E}_{D,x,I} \left[ -\log a_\theta(\tilde{\alpha}(D, x, I)|x, D) \right] \tag{9}$$
$$= \mathbb{E}_{D,x,\tilde{\alpha}} \left[ -\log a_\theta(\tilde{\alpha}|x, D) \right] \tag{10}$$
$$= \mathbb{E}_{D,x} \left[ \mathbb{E}_{\tilde{\alpha}}[-\log a_\theta(\tilde{\alpha}|x, D)] \right] \tag{11}$$
$$= \mathbb{E}_{D,x} \left[ \mathrm{CE}(p(\tilde{\alpha}|D, x), a_\theta(\tilde{\alpha}|x, D))] \right] \tag{12}$$
$$= \mathbb{E}_{D,x} \left[ \mathrm{KL}(p(\tilde{\alpha}|D, x), a_\theta(\tilde{\alpha}|x, D))] \right] + C, \tag{13}$$

where CE is cross-entropy and KL is the Kullback-Leibler divergence. This shows that our training loss optimizes for $a_\theta$ to approximate $p(\tilde{\alpha}|D, x)$ under the assumption that $q$ is exact. Now we can use $a_\theta$ in Equation 7 to approximate all ES variants, as $\mathbb{E}_{\tilde{\alpha} \sim a_\theta(\cdot|x,D)}[\tilde{\alpha}] \approx \mathbb{E}_{\tilde{\alpha} \sim p(\tilde{\alpha}|D,x)}[\tilde{\alpha}]$. □

## 5 EXPERIMENTAL SETUP

**In-Distribution Experiments.**    We replicate the first setup by Hvarfner et al. (2022) (JES) and consider their 2D setting and 6D setting with fixed hyperparameters and fixed lengthscales (See Table 3 in Appendix X for the exact prior). We train twelve $\alpha$-PFN models, one for each MES, PES and JES variants, one per dimensionality, and each variant is trained either with clustered data or without. Data is not normalized before feeding it into the transformer, not during training or during testing (BO). The PFN training set size is sampled uniformly in $[1, 50]$.

**Fully Bayesian for Out-of-Distribution Experiments.**    We train three fully Bayesian $\alpha$-PFNs, one per ES variant, on GP sample paths from the hyperprior with varying dimensionalities (1-6D). We use varying lengthscales per dimensionality, e.g. with automatic relevance determination. For all details regarding the prior see Table 3. Data is not normalized before feeding it into the transformer during training, but data is normalized during testing / BO: the training data is standardized using statistics of the training data itself. The PFN training set size is uniform on $[1, 50]$.

**Evaluation datasets.**    For Out-of-Distribution, we evaluate on well-known synthetic test functions for BBO. The Fully-Bayesian PFN has only been trained on dimensionalities up to 6D. We evaluate on real-world tasks provided by LC-Bench (Zimmer et al., 2021), which are 7D hyperparameter optimization tasks from MLP trainings on OpenML classification datasets (Vanschoren et al., 2014).

**Baselines.**    We compare the performance of our $\alpha$-PFNs with the existing Entropy Search approximations in the BoTorch library (Balandat et al., 2020): JES (Hvarfner et al., 2022), (GIBBON-)MES (Moss et al., 2021) and PES (Hernández-Lobato et al., 2014). For the in-distribution experiments, the priors are exactly the same. Since no fully Bayesian entropy search implementations exist, we compare against MCMC-ES instead, which approximates a posterior over the acquisition, rather than computing the acquisition of the fully Bayesian model. For our baselines we use NUTS (Hoffman et al., 2014), which uses Hamiltonian Monte Carlo (HMC), again making use of the BoTorch library for JES, GIBBON-MES and PES. Furthermore, we use Log EI, which is state-of-the-art more numerically stable variant of Expected Improvement (Ament et al., 2023). To evaluate the quality of our (unconditional) Base-PFN model, we reproduce the baseline of Müller et al. (2023) that performs EI with PFNs. For more details of evaluation see Appendix E.

## 6 RESULTS

We evaluate $\alpha$-PFN across progressively more challenging settings to assess its BO performance in terms of regret and speed. First, we test models trained and evaluated on fixed-hyperparameter

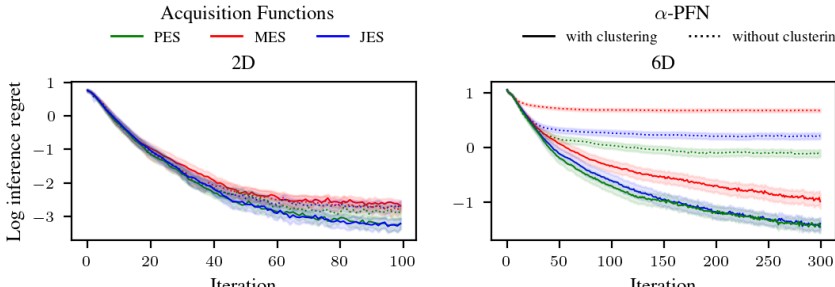

Figure 3: BO results on in-distribution data with fixed hyperparameters. In 6D, severe domain shifts occur when the PFN is trained on uniform, non-clustered data, but are reduced when trained on $x$'s clustered around local minima, resembling BO trajectories.

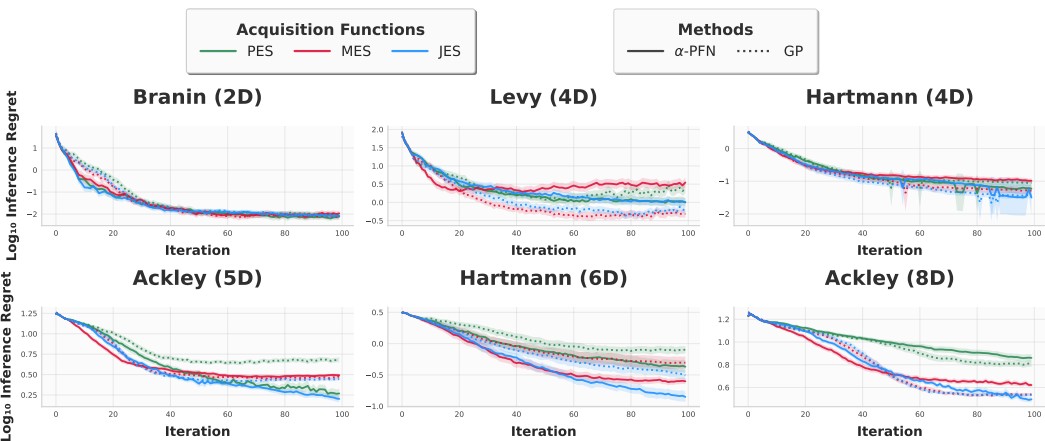

Figure 4: Inference regret for the GP-MCMC (NUTS) and $\alpha$-PFN. Shaded area indicates 2 SE.

GP priors to establish a clean in-distribution baseline in a fully controlled setting to evaluate the clustering. We assess a fully Bayesian $\alpha$-PFN trained on a GP hyperprior, evaluating it on synthetic functions and real-world BO problems from LC-bench, to measure out-of-distribution performance and comparing it with the GP performance. We also evaluate the speed ups.

**Fixed-Hyperparameters Results on the GP Prior (In-Distribution).** Figure 3 reports the log inference regret per iteration averaged across 100 GP function sample paths for the GP-based and PFN-based acquisition functions on samples of their 2D and 6D GP prior with fixed hyperparameter settings. Interestingly, the PES-PFN performs surprisingly well compared to the traditional PES approximation and is competitive to JES-PFN. This is surprising, since the PES approach of Hernández-Lobato et al. (2014) usually performs considerably worse than MES and JES with traditional approximations, as shown also by Hvarfner et al. (2022) and Wang and Jegelka (2017). This indicates that the PFN may have learned a better approximation to PES acquisition than the handcrafted approximation of Hernández-Lobato et al. (2014).

**Fully Bayesian Results on Synthetic Test Functions (Out-of-Distribution).** The results are in Figure 4, which are the result of 100 repititions The PFN oftens closely matches the performance. A major excpetion is Levy 4D, where the PFN falls significantly behind, and where PFN-MES inference regret increases, suggesting Levy is too far out of distribution. On Ackley 8D the PFN is ahead up to 50 iterations, afterward it starts to slightly lag in performance, suggesting that generalizing to both higher dimensions and higher iterations seen during training remains difficult. On Branin 2D, Ackley 5D, and Hartman 6D, the PFN shows significant performance improvements over the GPs. JES seems consistently the best, but the Ackley 5D result shows that the PES-PFN can obtain

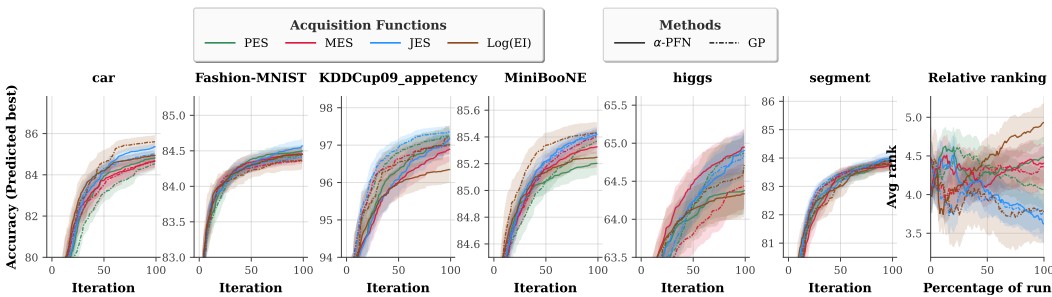

Figure 5: Results on LC-Bench. The shaded area indicates 2 standard errors.

excellent performances, which is surprising since ES literature finds the PES approximation to lead to suboptimal performance. This suggests $\alpha$-PFN-PES can learn good approximations.

Table 1: Runtime Analysis (minutes) by Problem Dimension. See Table 4 for details.

|  | 2D (Branin) | | | 4D (Levy, Hartmann) | | | 8D (Ackley) | | |
|---|---|---|---|---|---|---|---|---|---|
|  | GP-CPU | PFN-GPU | Speed up | GP-CPU | PFN-GPU | Speed up | GP-CPU | PFN-GPU | Speed up |
| MES | 13.30 | 1.41 | 9.43 | 22.74 | 2.57 | 8.8 | 53.71 | 2.20 | 24.41 |
| JES | 13.73 | 1.43 | 9.60 | 41.83 | 2.30 | 18.18 | 67.74 | 2.35 | 28.82 |
| PES | 14.83 | 1.54 | 9.62 | 42.64 | 2.68 | 15.91 | 38.96 | 2.15 | 18.12 |

**Timing Results.** The timing results are given in Table 4. Observe that the speed-ups are strongly dependent on the dimension and method. These runtimes are derrived from the synthetic test functions that adhere to the dimensionalities reported in the table and averaged. The speed up is computed by deviding the average runtime of the GP by the average runtime of the PFN-GPU. Observe that the PFN-GPU is consistently faster, and that speed ups are consistently larger than a factor of 8. The larger the dimension, the larger the speed-up.

**Fully Bayesian Results on Real-World LC-Bench HPO Tasks (Out-of-Distribution).** In our final experiment we consider optimizing real world black box optimization tasks from LC-Bench, see Figure 6. These are results of 100 randomly initialized runs. Performances are often quite close, which is also reflected in the average rankings (over problems) that are quite high (last panel). Notably are the huge improvement of MES-PFN or MES-GP for higgs. The KDDCup is challenging for all $\alpha$-PFNs, suggesting it is affected by a domain shift of the Base model. Given that the EI-PFN performs significantly worse than Log-EI, this seems likely. Otherwise, due to large standard errors, it is hard to draw strong conclusions regarding performance differences, besides that the $\alpha$-PFNs mostly perform on-par with the GPs.

## 7 SUMMARY, LIMITATIONS, AND FUTURE RESEARCH

Our results demonstrate that PFNs can be used for Entropy Search. We show that $\alpha$-PFN is capable of simulating the state-of-the-art (JES) at reduced runtimes. The strong speed improvements highlight its ability to learn more efficient approximations than handcrafted alternatives. Note that, it is difficult to compare runtimes, as the hyperparameters of the baselines, such as the number of MC samples used, can influence the runtimes significantly. We tried to set to reasonable values.

The $\alpha$-PFN often matches performance of the GPs, except for functions or datasets that are out of distribution. One way to mitigate this, is through developing more diverse priors or transformations at test time. A major strength of our framework is its flexibility. While we used a GP-based hyperprior to align with standard ES methods, the PFN approach is agnostic to the choice of prior, e.g., Bayesian neural networks, ensembles, etc. could be used easily. Exploring alternative priors is a promising direction for future work. One bottleneck is that the $\alpha$-PFN needs to be retrained for each prior; something that may be resolved by methods like Whittle et al. (2025). So far, we have

trained models up to 6 dimensions for 50 iterations, and find that the PFNs can generalize to higher dimensional problems or higher iterations. The work of Yu et al. (2025) illustrates that our architecture should scale to high-dimensional BO (up to 500D). However, this is a significant compute investment which we postpone to future work.

In summary, our results position PFNs as a promising and general tool for acquisition function amortization. They invite further work on scaling, generalization to other Monte Carlo acquisition functions (Balandat et al., 2020), a more efficient implementation, and broader applications.

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

## A    PROOF THAT THE BASE-PFN APPROXIMATES THE CONDITIONED PPD

The loss used for training the conditional base-PFN is:

$$l_\theta = \mathbb{E}_{D \cup \{x,y\} \cup I} \left[ -\log q_\theta(y|x, D, I) \right]$$

Here, the $I$ is information that can be conditioned on. This is either: $x^*$, $f^*$ or both: $(x^*, f^*)$. Note that during the training, we sample $D$, $(x, y)$ and $I$ jointly. In practice, this is done by sampling the Gaussian Processes using the Random Fourier Features approximation, where we can control the identity of the Gaussian Process by a latent vector $w$. For such a Gaussian Process, we can bruteforce compute the maximum and its location, resulting in the information $I$ that we store together with $w$.

**Insight 2.** *The proposed objective $l_\theta$ is equal to the expectation of the cross-entropy between the true conditional PPD and its approximation $q_\theta$.*

*Proof.* The proof is analogues to the proof by Müller et al. (2021), where we in addition condition the true posterior on $I$ and the neural network $q_\theta$.

$$l_\theta = -\int_{D,x,y,I} p(x,y,D,I) \log q_\theta(y|x,D,I) = -\int_{D,x,I} p(x,D,I) \int_y p(y|x,D,I) \log q_\theta(y|x,D,I)$$

$$= \int_{D,x,I} p(x,D) H(p(y|x,D,I), q_\theta(y|x,D,I) = \mathbb{E}_{x,D,I \sim p(D)} \left[ H(p(y|x,D,I), q_\theta(y|x,D,I)) \right]$$

where $H(p,q)$ indicates the cross entropy between $p$ and $q$. $\qquad\qquad\square$

## B    PRE-COMPUTING APPROXIMATE GP DATA AND THEIR MAXIMA

### B.1    GP APPROXIMATION

The Random Fourier Feature (RFF) approximation obtains a sample path, which correspond to a realization of a Gaussian Process. Such sample paths are represented by a weight vector $w$ in RFF space. To obtain a sample path, one samples $w$. The distribution of $w$ is determined by the kernel hyperparameters. Now, after fixing $w$, it is possible to evaluate the sample path at any position $x$. For a fixed $w$, it is thus possible to maximize it over the domain to estimate $x^*$ and $f^*$. Note that this is a difficult global optimization problem, which we solve approximately by an ensemble of optimizers that are restarted multiple times. Note that, the extension to the Fully Bayesian GP is natural. For the Fully Bayesian case we have a two-stage procedure: (1) we sample the kernel's hyperparameters, which determines the distribution over $w$. (2) we sample $w$ from this distribution.

For each setting, we generate millions of approximate samples from the corresponding GP prior (see Table 3) using the Random Fourier Feature (RFF) approximation Rahimi and Recht, 2007. We use 5000 RFFs when computing the GP approximations for dataset sizes up to 10 million, and 500 RFFs otherwise.

### B.2    IDENTIFYING APPROXIMATE GP MAXIMIZERS

Computing the maximum of GP sample paths drawn via RFFs is a non-convex optimization problem. To identify the maximizers, we perform random search followed by first-order gradient-based optimization. First, we describe the optimizer and its hyperparameters, and afterward we describe the ensemble construction. We use either SGD or Adam and a batch of size `num_samples` points over the GP in parallel. The initialization is done either uniformly at random over the domain $[0,1]^d$ (`resample_init` is False) or by trying a large number of points (`num_repeats` times `num_samples` points are tried), computing their function values, and keeping the best `num_samples` (if `resample_init` is True). Each optimizer is run for `n_iterations_max` iterations. During the optimization, we monitor the current best seen GP value so far and store it. If the current best point does not improve (compared with a tolerance `tol`), we increase a counter indicating the patience, and otherwise the patience counter is reset. After the patience counter reaches a value of `patience`, we decay the learning rate by a factor of `decay_factor`. If the learning rate is decayed more than `max_decays` times, we stop the optimization early. One should take care with points that move outside the domain during optimization, as optima are often located at

Table 2: Hyperparameter grid values for building the GP Maximizer Ensemble.

| hyperparameter | grid values | hyperparameter | grid values |
| --- | --- | --- | --- |
| adam | [True, False] | max_decays | [1, 5, 10, 15] |
| init_lr | [0.001, 0.01, 0.1, 1] | tol | [1e-1, 1e-3, 1e-6] |
| resample_init | [True, False] | decay_factor | [0.1, 0.5, 0.99, 1] |
| num_samples | [50,200,1000] | clamp | [True, False] |
| n_iterations_max | [10, 100, 1000] | patience | [1, 5, 10, 20] |
| num_repeats | [10, 100, 1000] | | |

the edge, especially for higher length scales / dimensions. If `clamp` is active, we always move the points back inside the domain by clamping. If `clamp` is not true, we randomly initialize these points. If not specified, we use the default values for the optimizers as specified in PyTorch version 2.3.1.

### B.3 MAXIMIZER ENSEMBLE CONSTRUCTION

We use the first 1000 GP sample paths to build the optimizer ensemble. We build an candidate set of a 1000 optimizers with hyperparameters sampled from the grid as defined in Table 2. Because we want to determine the maxima for a large number of sample paths, we need to design a small ensemble with a low runtime yet good performance. We measure performance in terms of the regret of the ensemble, that is, the performance of the ensemble compared to the performance of the best optimizer in the candidate set. To reliably estimate the ensemble performance, we use 5-fold cross validation, where a train fold is used for the ensemble building, and the test fold is used for ensemble evaluation. The ensemble is constructed greedily by forward selection, where only candidates are considered that have an average runtime of less than 10 seconds per GP optimization. We keep adding ensemble members until the regret indicates we find the optimum with an approximate precision of 1e-6. We merge the ensembles together over the different training folds to come to a final ensemble. Note that the ensemble is build separately for each prior of Table 3.

## C   BASE PFN TRAINING DETAILS

We closely follow the original PFN architecture and training pipeline, used in previous works (Müller et al., 2021; 2023). We use a small ($\pm$15M parameter) decoder-only transformer, with 6 layers, each using an embedding size of 128, 4 attention heads, and 1024 units in the hidden expansion layer. We use the PFN regression head proposed by Müller et al. (2021) to model the output distribution. The output distribution is the Full-Bar Distribution (also called Riemannian distribution) with Full-Support. We use 5000 bins and determine the bin size such that each bin contains roughly as many of the training targets.

We minimize the cross-entropy loss, using AdamW with a batch size of 100 datasets, a cosine decay learning rate schedule with maximum 0.0001, and linear warmup over the first 25% iterations of the training run. Our models for the fixed hyperparameter settings were trained on 60M datasets, 90M were used for the fully Bayesian model. The GP sample used for a dataset is selected randomly from the pregenerated samples (see Appendix B), 10M for the fixed hyperparameter and 90M for the fully Bayesian setting, and is possibly reused multiple times. To counteract overfitting, and to encourage symmetry, we perform mirror and reflection augmentations on the domain $A$. The context size $C$ (train split) is fixed per batch and sampled from an exponentially-biased distribution favoring shorter contexts, with probabilities proportional to $0.99^c$ for $c \in [0, \ 50 \cdot \ D - 1]$. The context and query points are chosen as described in Appendix F.

## D   $\alpha$-PFN TRAINING DETAILS

The PFN model directly predicting the (expected) information gain closely follows the architecture and training of the base model. The main difference is that no conditioning tokens were used (and it does not take $I$ as input), and that the prediction targets for queries are not $y$, but the oracle

| Priors | d | $\sigma_f$ | $\sigma_l$ | $\sigma_n$ | $\mu$ |
|---|---|---|---|---|---|
| 1. Fixed Hyperparameters 2D | 2 | $\sqrt{10}$ | 0.1 | 0.1 | 0 |
| 2. Fixed Hyperparameters 6D | 6 | $\sqrt{10}$ | 0.3 | 0.1 | 0 |
| 2. Fully Bayesian | [1,6] | 1 | $P(D)$ | LN(-4,1) | N(0, 0.5) |

Table 3: The different priors on which we train the base-models and the $\alpha$-PFN. $d$ indicates the dimension of the input space. We use the squared exponential kernel, where $\sigma_f^2$ is the variance of the Gaussian Process (output scale) and $\sigma_l$ is the length scale of the kernel. The lengthscale per dimension is sampled for fully Bayesian (corresponding to Automatic Relevance Determination), where the sampling is done from $P(D) = LN(\mu_0 + \frac{logD}{2}, \sigma_0)$ with $\mu_0 = -0.75$ and $\sigma_0 = 0.75$ (this prior was inspired by Hvarfner et al. (2024)). We add zero mean Gaussian noise $N(0, \sigma_n)$ to the GP, and we set the mean $\mu$ of the GP to zero, except in the Fully Bayesian case where the mean is sampled from $N(0, 0.25)$ per task.

information gain from equation 5. Note that we trained a total of 9 models, one for each setting and ES variant, on 10M datasets for the fixed hyperparameter and 20M datasets for the fully Bayesian settings. For all models, we use 1000 bins determined such that roughly equally many of the training targets fall in each bin. Furthermore, we use the full-support output head (Müller et al., 2021), where the outmost "bins" are open and modeled as half-normals.

# E   BAYESIAN OPTIMIZATION EXPERIMENT DETAILS

The initial design consists of $d$ uniformly sampled points. At each BO iteration, the acquisition function is optimized using Botorch routine `optimize_acqf`, with the following hyperparameters: 1024 uniform points as initial candidates, up to 25 gradient steps, a batch size of 128, and 8 restarts. We follow a similar optimization procedure to compute the maximizer of predictive posterior distribution, required for computing the inference regret.

We evaluate all methods in terms of inference regret for synthetic tasks, which is the standard evaluation measure for methods using information theoretic acquisition functions. Inference regret is defined as $f(x^*) - f(\hat{x}^*)$, where $\hat{x}^*$ is the maximizer of the posterior predictive distribution, i.e., $\hat{x}^* = \mathrm{argmax}_{x \in A} \mathbb{E}[q(y|D, x)]$. Note that this maximizer is approximated by performing gradient descent on $q(y|D, x)$, which in this case can either be the surrogate GP or the PFN base-model, with a setup similar to that considered for the acquisition function optimization (see Appendix E). For acquisition function optimization we use the standard Botorch routine for GPs, but for the PFN we deviate from this and use a new simpler optimizer. For real-world tasks we report $f(\hat{x}^*)$ (similar to inference regret). For evaluation we use 100 iterations, while PFNs are trained for 50.

# F   SYNTHETIC OPTIMIZATION TRACE GENERATION

## F.1   MOTIVATION

Previous work (Müller et al., 2021; 2023; Rakotoarison et al., 2024) considered context and query points sampled uniformly. However, in real Bayesian Optimization (BO) traces, the context points follow a structured search pattern, dynamically balancing global and local search and often forming clusters around local optima. Likewise, uniform query points, in high dimensions, are unlikely to be near any of the context points, limiting the opportunity to learn to exploit the information they provide. Ideally, we would use actual optimization traces from the Entropy Search BO procedures. However, this would introduce a dependency on our surrogate model, leading to a chicken-and-egg problem. Alternatively, using BoTorch traces would restrict ourselves to the GP priors it supports. Instead, we propose a simple and efficient synthetic procedure that generates context and query points in a manner that mimics real BO traces.

## F.2 PROCEDURE

Our synthetic optimization trace generation procedure, detailed in Algorithm 1, aims to replicate the characteristics of real BO traces by blending global and local search. The key components are:

- **Global search**: Points are sampled uniformly at random within the search space, with an additional probability $\epsilon$ of selecting a point exactly on the edge. This helps in exploring boundary effects which are important in higher dimensions as the optimizer increasingly often lies exactly on the edge.

- **Local search**: The next context points are drawn from a Gaussian distribution centered on the best observed context point so far. For query points, we select an arbitrary context point as the center. If the optimizer $x^*$ is provided, it is sometimes chosen as the center, which facilitates learning the effect of conditioning on $x^*$.

- **Dynamic search adaptation**: BO dynamically transitions from global to local search over time. To model this, we define a local search probability $\alpha_i$ that linearly decreases over $L$ steps. This ensures that earlier points explore the space globally, while later points refine the search locally.

- **Avoiding duplicate points**: Though not explicitly shown in the pseudocode, we ensure that no duplicate points occur in the trace. This frequently happens in corner regions. If a newly generated point coincides with an existing corner point, it is resampled.

This procedure effectively balances exploration and exploitation, producing synthetic traces that resemble real BO optimization trajectories while remaining computationally efficient.

## G TRAINING COMPUTE RESOURCES

Pre-computing the GP sample paths and their maximizer (Appendix B) took approximately 40K CPU hours per setting. Training the base PFN models (Appendix C) took 15, 40, and 70 hours on NVIDIA L40S for settings 1, 2, and 3 respectively. Training one of the three $\alpha$-PFN models (Appendix D) took 3, 6, and 12 hours on NVIDIA L40S for settings 1, 2, and 3 respectively.

## H BROADER IMPACT STATEMENT

This work makes a fundamental contribution to the development of efficient acquisition functions for Bayesian optimization by replacing costly sampling-based approximations with learned approximations via Prior-data Fitted Networks (PFNs). As such, it does not directly interact with sensitive application areas or decision-making domains and is unlikely to pose immediate negative societal risks. By significantly reducing the computational overhead of Entropy Search, our approach lowers the resource requirements for effective Bayesian optimization. This has two positive implications: (1) it supports the democratization of advanced black-box optimization methods, enabling broader access to state-of-the-art tools without requiring large compute budgets; and (2) it contributes to reducing energy consumption in hyperparameter optimization and similar tasks, offering a modest but meaningful environmental benefit.

## I CODE AND MODEL CHECKPOINTS

To support reproducibility and transparency, we provide an anonymous online repository containing the complete codebase used in this work. This includes code for training, evaluation, data preprocessing, and pretrained model checkpoints. Instructions for setup and usage are provided in the README file. The repository can be accessed at the following link: `https://anonymous.4open.science/r/pfns4es-C6B2/readme.md`. The link is anonymized in accordance with the double-blind review process.

---

**Algorithm 1** Generate Optimization Trace

---

1: **Inputs:**
2:    $L$: length of the trace
3:    $C$: number of context points (i.e., we have $L - C$ query points)
4:    $d$: dimension of search space
5:    $GP\_sample$: function to evaluate GP values
6: **Outputs:**
7:    $trace$: matrix of shape $(L, d)$ with context/query points in the trace
8:    $y$: vector of length $C$ with function values for context points
9: **Procedure:**
10: Initialize $trace \leftarrow$ zero matrix of size $(L, d)$
11: Initialize $y \leftarrow$ zero vector of size $C$
12: $\epsilon = (1 - u^{\frac{d}{6}})$ with $u \sim U(0, 1)$                             ▷ Sample edge probability
13: $\sigma \sim \text{LogNormal}(-3, 0.5)$                         ▷ Sample local search step size
14: Sample initial / final local search probability:
15:    $\alpha_0 = \min(v_1, v_2, v_3)$
16:    $\alpha_L = \max(v_1, v_2, v_3)$
17:    with $v_1, v_2, v_3 \sim U(0, 1)$
18: Start trace from a random point in search space:
19:    $best\_point = trace[0] \leftarrow clip(w, 0, 1)$ with $w \sim U^d(-\frac{\epsilon}{2}, 1 + \frac{\epsilon}{2})$
20:    $y_{best} = y[0] \leftarrow GP\_sample(trace[0])$
21: **for** $i = 1$ to $L - 1$ **do**
22:    $\alpha_i \leftarrow \alpha_0 + (\alpha_L - \alpha_0) \cdot (i/L)$          ▷ Determine local search probability
23:    $local \leftarrow \text{Bernoulli}(\alpha)$              ▷ Determine local or global search
24:    **if** $local$ **then**
25:       **if** $i < C$ **then**
26:          $inc \leftarrow best\_point$             ▷ Sample near the best point thus far
27:       **else**
28:          Choose $inc$ randomly from context points ($trace[: C]$)
29:       **end if**
30:       $trace[i] \leftarrow clip(x, 0, 1)$ with $x \sim \mathcal{N}^d(inc, \sigma^2)$
31:    **else**
32:       $trace[i] \leftarrow clip(x, 0, 1)$ with $x \sim U^d(-\frac{\epsilon}{2}, 1 + \frac{\epsilon}{2})$    ▷ Global search
33:    **end if**
34:    **if** $i < C$ **then**            ▷ For context point, sample value and update best
35:       $y[i] \leftarrow GP\_sample(trace[i])$
36:       **if** $y[i] > y_{best}$ **then**
37:          $best\_point \leftarrow trace[i]$
38:          $y_{best} \leftarrow y[i]$
39:       **end if**
40:    **end if**
41: **end for**
42: **return** $trace, y$

---

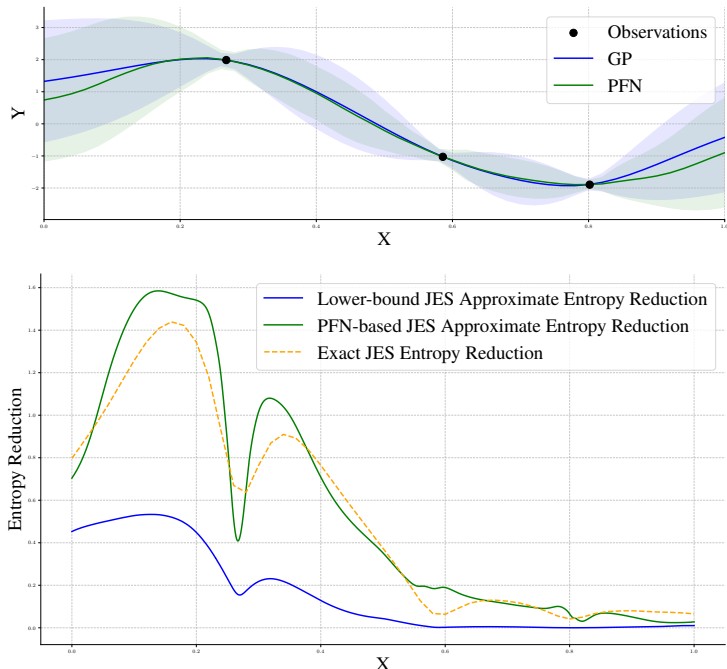

Figure 6: Entropy reduction comparison. The exact JES entropy reduction is computed using rejection sampling approach introduced in Hernández-Lobato et al. (2014).

## J  DETAILED RUNTIME

Table 4 details the overall BO, the model fitting, and acquisition function optimization runtimes for GP and PFN.

Table 4: Detailed cumulative runtime (minutes) of 100 BO iterations for PFN and GP, on CPU and GPU. Table covers runtime for model fitting, acquisition optimization, and posterior predictive distribution optimization for inference regret computation.

| | | Model Fit | | | Acq. Opt. | | | PPD Opt. | | | Total | | |
|---|---|---|---|---|---|---|---|---|---|---|---|---|---|
| | | MES | JES | PES | MES | JES | PES | MES | JES | PES | MES | JES | PES |
| **2D** | GP-CPU | 6.16 | 5.11 | 5.41 | 6.835 | 8.386 | 9.098 | 0.30 | 0.24 | 0.32 | **13.30** | **13.73** | **14.83** |
| | GP-GPU | 6.04 | 6.05 | 5.98 | 0.433 | 0.551 | 1.173 | 0.22 | 0.37 | 0.22 | **6.69** | **6.97** | **7.38** |
| | PFN-CPU | 0.0 | 0.0 | 0.0 | 2.35 | 3.20 | 2.10 | 3.26 | 3.71 | 2.92 | **5.61** | **6.91** | **5.02** |
| | PFN-GPU | 0.0 | 0.0 | 0.0 | 0.64 | 0.68 | 0.78 | 0.78 | 0.74 | 0.77 | **1.41** | **1.43** | **1.54** |
| **4D** | GP-CPU | 6.17 | 7.56 | 9.27 | 16.094 | 33.597 | 32.789 | 0.48 | 0.68 | 0.58 | **22.74** | **41.83** | **42.64** |
| | GP-GPU | 6.84 | 7.10 | 7.10 | 0.903 | 1.067 | 1.769 | 0.38 | 0.42 | 0.41 | **8.12** | **8.59** | **9.28** |
| | PFN-CPU | 0.0 | 0.0 | 0.0 | 3.41 | 6.37 | 4.32 | 4.90 | 7.77 | 4.98 | **8.31** | **14.13** | **9.30** |
| | PFN-GPU | 0.0 | 0.0 | 0.0 | 1.47 | 1.26 | 1.36 | 1.10 | 1.04 | 1.33 | **2.57** | **2.30** | **2.68** |
| **8D** | GP-CPU | 8.75 | 7.59 | 7.93 | 44.076 | 59.245 | 30.095 | 0.89 | 0.91 | 0.93 | **53.71** | **67.74** | **38.96** |
| | GP-GPU | 8.17 | 8.03 | 8.29 | 1.567 | 1.386 | 1.974 | 1.00 | 0.88 | 0.87 | **10.73** | **10.30** | **11.14** |
| | PFN-CPU | 0.0 | 0.0 | 0.0 | 8.90 | 7.55 | 9.13 | 8.80 | 8.24 | 9.02 | **17.70** | **15.79** | **18.15** |
| | PFN-GPU | 0.0 | 0.0 | 0.0 | 1.30 | 1.33 | 1.19 | 0.90 | 1.02 | 0.97 | **2.20** | **2.35** | **2.15** |

## K  ENTROPY REDUCTION COMPARISON

Figure 6 provides a qualitative comparison of the approximation of entropy reduction with GP and PFN. The models use the same fully Bayesian prior as defined in Table 3. The figure shows the merits of the PFN approximation compared to GP.

