# OpenReview forum: "$\alpha$-PFN: Fast Entropy Search via In-Context Learning"
_ICLR.cc/2026/Conference — Submitted to ICLR 2026_

### Official Review · Reviewer_PgkG · 2025-10-27

**Soundness:** 3
**Presentation:** 3
**Contribution:** 3
**Rating:** 6
**Confidence:** 2

**Summary:**

- This paper addresses Bayesian optimization (BO) with information-theoretic acquisition functions (ACQFs) such as Entropy Search (ES), Predictive Entropy Search (PES), Max-value Entropy Search (MES), and Joint Entropy Search (JES).
- These ACQFs lack analytical closed forms and require costly Monte Carlo–based approximations, which make them computationally expensive. This motivates the use of approximations for these ACQFs.
- The authors propose a two-stage amortization framework using Prior-data Fitted Networks (PFNs) to approximate these ACQFs.
- The first-stage PFN (base PFN) is trained on millions of simulated Gaussian Process (GP) sample paths and their optima. It predicts the posterior predictive distribution $q(y_{tst}|D_{trn},x_{tst},I)$, conditioned on information about the true optimum ($I=x^∗,f^∗,(x^∗,f^∗)$ depending on the ACQF variant). The entropy difference between the conditioned and unconditioned predictions provides the information gain, which serves as training data for the next stage PFN.
- The second-stage PFN ($\alpha$-PFN) is then trained to directly predict the expected information gain (the ACQF value), effectively learning to approximate PES, MES, and JES in a single forward pass
- During actual BO loop, only the $\alpha$-PFN is used to evaluate the acquisition function and select the next point.
- Experiments on synthetic and real-world benchmarks show that $\alpha$-PFN achieves performance competitive with standard GP-based PES/MES/JES, while providing substantial computational speedups up to 30x faster in some of the test problems.

**Strengths:**

- The paper proposes an original two-stage PFN framework to efficiently approximate expensive information-theoretic ACQFs in BO. The idea of amortizing PES/MES/JES computation via neural networks is creative and addresses a key bottleneck in practical BO.
- Theoretical analysis is rigorous and clearly connects the two PFN stages, demonstrating that the $\alpha$-PFN can recover the expected information gain under mild assumptions.
- Moreover, the graphical illustrations and diagrams are clear and informative, effectively conveying the high-level ideas and overall workflow of the proposed framework.

**Weaknesses:**

- The paper motivates its contribution by the high computational cost of information-theoretic ACQFs, yet it does not clearly articulate why such ACQFs are preferable to simpler and cheaper alternatives like Expected Improvement (EI) or Upper Confidence Bound (UCB). Without this justification, it is unclear when the proposed acceleration would be practically valuable. A stronger discussion or empirical comparison highlighting scenarios where PES/MES/JES outperform EI would improve the motivation and significance of the work.
- Although inference is much faster, training remains highly resource-intensive, requiring millions of GP simulations and two large PFN models. This appears to be a major limitation of the approach and should be emphasized more clearly in the main text.
- Furthermore, the reported performance–runtime comparison could be presented in a more informative manner, for example by including a Pareto front that visualizes the trade-off between optimization performance and computational cost.
- This analysis should also account for the overall runtime, including the upfront cost of training the α-PFN.

**Questions:**

- Line 168: Typo $D_{trn}$
- What is the dimension of each LC-Bench problem? Having parenthesis in the plot title as in synthetic test functions would be helpful.
- How $\alpha$-PFN based acquisition function is optimized in each iteration?
- It would be great to see numerical experiments in higher dimension to see how the performance of the proposed method behaves.

---

> ### Author Response · Authors · 2025-12-03
>
> We thank the reviewer for their feedback, and appreciate the questions regarding the relevance of our method as well as its cost. Below, we have added all the relevant information regarding the cost of our method, and will add it
>
> **Why Entropy Search?** We agree with the reviewer on this point - there are cheap and practically strong BO methods, and GP-based entropy search is not cheap. However, Entropy Search can certainly be both strong and practically versatile given the plethora of available MO/MF-variants [5-12]. Thus, we think it could be highly impactful to close the outstanding gaps with regard to complexity and computational expense.
>
> **Highlight the resource-intensive training procedure more.**  We agree that this is a major limitation and we will emphasize it further. We mention this already in the introduction and conclusion of our paper, however, especially in the conclusion, we agree this is not clearly emphasized enough. We will rewrite it as follows: “One bottleneck is the large upfront computational cost of training the Alpha-PFN, which needs to be redone for each prior; something that may be resolved by…”. We will also add a remark to the method section to highlight more clearly the expensive training costs as well.
>
> **The up-front training costs should be incorporated in the runtime comparison.** We think that maybe one insight is missed here by the reviewer: training needs to be done upfront once, but after this, the model can be reused indefinitely. For example, upon publication, we will openly share our model weights, allowing anyone to use our model for BO without any upfront training costs. Especially in the context of Green-AI; this makes a lot of sense: these computations are done once, and are therefore amortized for all future users.
> Let us illustrate with a (simplified) computation. Assuming the 2D setting with PES. We have that using the classical variant takes 11 minutes. Our Alpha-PFN takes 41 seconds. However, training the Alpha-PFN costs 40k CPU hours and 80 GPU hours. Let us calculate in hours for simplicity. Thus the compute costs are approximately 41k hours.
> Let us assume we have 1000 users that each perform 1000 BO experiments. Using the Alpha-PFN approach; the compute time would be 1.3 years. Meanwhile, using the traditional GP approach would take 20.9 years. Our precomputation step took 4.6 years of compute. This illustrates the benefit of ammortization: our compute cost is carried out once, but all users of our approach in the future will have reduced computational cost. This leads to two advantages: less compute use, and less climate impact of BO-experiments on the long term.
> Note that we don’t have the illusion that we will reach such a large audience of users, but we believe future iterations of this work down the line could have such an impact.
>
> **Dimension of the LC-Bench functions.** LCbench is a 7D benchmark. It includes MLP hyperparameters: batch_size (int), max_dropout (float), max\_units (int), num_layers (int), learning\_rate (float), momentum (float), weight_decay (float).
>
> **How is the acquisition optimized for the Alpha-PFN?** Please see Appendix E: “The initial design consists of d uniformly sampled points. At each BO iteration, the acquisition
> function is optimized using Botorch routine optimize acqf, with the following hyperparameters: 1024 uniform points as initial candidates, up to 25 gradient steps, a batch size of 128, and 8 restarts. To clarify further: 8 random restarts means that 8 candidates are optimized with gradient descent. Initially, the 1024 uniform points are evaluated, and only the 8 with the best acquisition values are kept for further optimization with gradient descent.
>
>
> **Need for higher-dimensional experiments.** First, we would like to stress that many relevant HPO problems are often low dimensional (e.g., HPO) and evaluations up to 8 dimensions are hardly uncommon in BO literature. High dimensional BO is an active research area, presenting many challenges. Most of these challenges are orthogonal to the main contribution of our work, e.g., the search procedure and prior would need to be adapted accordingly. Since training models to support high-dimensional settings would be computationally very demanding, we have decided not to carry this out in the limited time of the rebuttal - we postpone this to future work. We anticipate that further scaling may be non-trivial, this primarily because of computational and practical issues - not due to some intrinsic limitation of our approach, e.g., [11] shows BO using a (Tab)PFN-based model performing very strongly up to 500 dimensions. More generally, the latest TabPFN model [12] supports up to 2000 dimensions.
>
> ______________
>
> We once again thank the reivewer for their feedback, and hope that our motivation and clarification of the computational cost has improved the reviewer's perception of our paper.

---

> ### Author Response · Authors · 2025-12-03
> **References**
>
> [1] Ament, S., Daulton, S., Eriksson, D., Balandat, M., & Bakshy, E. (2023). Unexpected improvements to expected improvement for bayesian optimization. Advances in Neural Information Processing Systems, 36, 20577-20612.
>
> [2] Hernández-Lobato, J. M., Gelbart, M., Hoffman, M., Adams, R., & Ghahramani, Z. (2015, June). Predictive entropy search for Bayesian optimization with unknown constraints. In International conference on machine learning (pp. 1699-1707). PMLR.
>
> [3] Perrone, V., Shcherbatyi, I., Jenatton, R., Archambeau, C., & Seeger, M. (2019). Constrained Bayesian optimization with max-value entropy search. arXiv preprint arXiv:1910.07003.
>
> [4] Hernández-Lobato, D., Hernandez-Lobato, J., Shah, A., & Adams, R. (2016, June). Predictive entropy search for multi-objective bayesian optimization. In International conference on machine learning (pp. 1492-1501). PMLR.
>
> [5] Takeno, S., Tamura, T., Shitara, K., & Karasuyama, M. (2022, June). Sequential and parallel constrained max-value entropy search via information lower bound. In International Conference on Machine Learning (pp. 20960-20986). PMLR.
>
> [6] Suzuki, S., Takeno, S., Tamura, T., Shitara, K., & Karasuyama, M. (2020, November). Multi-objective Bayesian optimization using Pareto-frontier entropy. In International conference on machine learning (pp. 9279-9288). PMLR.
>
> [7] Tu, B., Gandy, A., Kantas, N., & Shafei, B. (2022). Joint entropy search for multi-objective Bayesian optimization. Advances in Neural Information Processing Systems, 35, 9922-9938.
>
> [8] Fernández-Sánchez, D., Garrido-Merchán, E. C., & Hernández-Lobato, D. (2020). Improved max-value entropy search for multi-objective bayesian optimization with constraints. arXiv preprint arXiv:2011.01150.
>
> [9] Garrido-Merchán, E. C., Fernández-Sánchez, D., & Hernández-Lobato, D. (2023). Parallel predictive entropy search for multi-objective Bayesian optimization with constraints applied to the tuning of machine learning algorithms. Expert Systems with Applications, 215, 119328.
>
> [10] Takeno, S., Fukuoka, H., Tsukada, Y., Koyama, T., Shiga, M., Takeuchi, I., & Karasuyama, M. (2020, November). Multi-fidelity Bayesian optimization with max-value entropy search and its parallelization. In International Conference on Machine Learning (pp. 9334-9345). PMLR.
>
> [11] Yu, R. T. Y., Picard, C., & Ahmed, F. (2025). GIT-BO: Hig-Dimensional Bayesian Optimization with Tabular Foundation Models. arXiv preprint arXiv:2505.20685.
>
> [12] Grinsztajn, L., Flöge, K., Key, O., Birkel, F., Jund, P., Roof, B., ... & Hutter, F. (2025). TabPFN-2.5: Advancing the State of the Art in Tabular Foundation Models. arXiv preprint arXiv:2511.08667.

---

### Official Review · Reviewer_Udjn · 2025-10-28

**Soundness:** 3
**Presentation:** 3
**Contribution:** 2
**Rating:** 4
**Confidence:** 3

**Summary:**

This paper challenges the speed up of the information-theoretic Bayesian optimization (BO) algorithms leveraging prior-data fitted networks (PFNs). Different from the existing studies that leverage PFNs for Bayesian predictions, this paper leverages PFNs to directly meta-learn the BO optimization trace. As a result, the authors report a consistent 12x speedup compared with the usual Gaussian process-based entropy search algorithms.

**Strengths:**

This paper is clearly well-written and interesting as a practical verification, leveraging the transformer for BO.
The speed up can be a solid benefit, particularly when the objective function evaluation cost is moderate.

**Weaknesses:**

- Since this paper concentrates on the experimental validation of the transformer-based network on the black-box optimization problem, I would expect more extensive experiments. In particular, when the evaluation cost of the objective function is moderate, I think that other black-box optimization methods, such as genetic algorithms, can be an option. Thus, the experimental result comparing such wider black-box optimization methods on the wall-clock time can be of interest.


- This paper concentrates on entropy search algorithms. On the other hand, in the widely studied optimization problems, such as vanilla, multi-objective, and multi-fidelity optimization, there are already cheap and practically strong BO methods. The benefit or future potential of the proposed method compared with such known cheap baselines was unclear to me.


- I think that one of the natural directions to leverage the neural network for bandit or optimization is employing neural network-based surrogate models. Indeed, there are several studies, such as neural bandit [1]. However, there is no discussion and comparison of studies that leverage neural networks in other ways.

[1] Dongruo Zhou, Lihong Li, Quanquan Gu, Neural Contextual Bandits with UCB-based Exploration, ICML2020.


Minor:
- The standard errors in the regret plot are heavily overlapped, which degrades the visibility. Please consider using an error bar or other change to improve the visibility.

**Questions:**

Please answer the above comments.

---

> ### Author Response · Authors · 2025-12-03
>
> Thank you for your review. Below we respond to the weaknesses regarding our exclusion of the requested baselines, and motivate our choice of focusing on the Entropy Search acquisition function class.
>
> **More blackbox baselines such as genetic algorithms.** A practical issue, in terms of fair comparison with BB methods, is that in Entropy Search, the evaluation is typically done in terms of Inference Regret (PES, MES, JES). Inference regret computes the best guess of the optima according to the surrogate model, and computes the regret (performance gap to the ground truth optimal point) with respect to this candidate. Other approaches, such as genetic algorithms, do not build such a surrogate model, and therefore don’t fit into our evaluation framework where we use Inference Regret.
>
> More generally, the central premise of our paper is improving upon existing entropy search methods, our work includes the required baselines to enable an apples-to-apples comparison. These baselines included are highly optimized ES implementations. We do not claim these are the best possible BO methods - let alone the best black box optimizer. In fact, some features of other BB methods (orthogonal to our contribution) are likely to give them an edge in some settings. We’d like to point out that Bayesian Optimization is the dominant approach to black-box optimization generally, typically outcompeting evolutionary approaches, see [1, 2]. As an example, CMA-ES is such an evolutionary approach, and is known to be outperformed by Bayesian Optimization methods with GPs (see, for example, [2]). The baseline that we compare to, Log EI (a more numerically stable variant of EI), has shown to be State-of-the-Art for black-box optimization [3]. We show that our Alpha-PFN is competitive with Log EI; and thus, we believe, already offers solid performance.
>
> **Why Entropy Search?** We agree with the reviewer on this point - there are cheap and practically strong BO methods, and GP-based entropy search is not always cheap. However, it can certainly be both strong and practically versatile given the plethora of available MO/MF-variants [5-12]. Thus, we think it could be highly impactful to close the outstanding gaps with regard to complexity and computational expense.
>
>
> **Regarding Weakness 3: Neural Bandit.** We agree that there is some similarity between our work and Neural Bandit, since both use neural networks and surrogate modeling, and therefore we will include it in related work. However, we believe that its not possible to use Neural Bandit for BO. See below.
> One might attempt to discretize the BO domain and cast it as a bandit problem to apply NeuralUCB, but this reduction is not meaningful. In contextual bandits, the optimal action typically depends on a context vector that changes every iteration, so the optimal action is time-varying. In BO, however, we search for a single fixed maximizer of an unknown function. Thus, the concept of an “action” in the bandit setting is fundamentally different from a query point in BO.
> These frameworks only coincide under restrictive assumptions—namely, a fixed context and an infinite action set. Even under this limit, NeuralUCB cannot be applied to BO in practice: action selection in Algorithm 1 requires enumerating all arms, which is infeasible in a continuous domain. Moreover, its regret guarantee scales as TK\sqrt{T K}TK​; as K→∞K \to \inftyK→∞, thus the guarantee becomes vacuous.
>
>
> ____________________
>
>
> Altogether, we hope these clarifications address the reviewer's concerns and strengthen the motivation and scope of our paper. Thank you again for your valuable input.

---

> ### Author Response · Authors · 2025-12-03
> **References**
>
> [1] Turner, R., Eriksson, D., McCourt, M., Kiili, J., Laaksonen, E., Xu, Z., & Guyon, I. (2021, August). Bayesian optimization is superior to random search for machine learning hyperparameter tuning: Analysis of the black-box optimization challenge 2020. In NeurIPS 2020 competition and demonstration track (pp. 3-26). PMLR.
>
> [2] Hvarfner, C., Hellsten, E. O., & Nardi, L. (2024). Vanilla Bayesian optimization performs great in high dimensions. arXiv preprint arXiv:2402.02229.
>
> [3] Ament, S., Daulton, S., Eriksson, D., Balandat, M., & Bakshy, E. (2023). Unexpected improvements to expected improvement for bayesian optimization. Advances in Neural Information Processing Systems, 36, 20577-20612.
>
> [4] Hernández-Lobato, J. M., Gelbart, M., Hoffman, M., Adams, R., & Ghahramani, Z. (2015, June). Predictive entropy search for Bayesian optimization with unknown constraints. In International conference on machine learning (pp. 1699-1707). PMLR.
>
> [5] Perrone, V., Shcherbatyi, I., Jenatton, R., Archambeau, C., & Seeger, M. (2019). Constrained Bayesian optimization with max-value entropy search. arXiv preprint arXiv:1910.07003.
>
> [6] Hernández-Lobato, D., Hernandez-Lobato, J., Shah, A., & Adams, R. (2016, June). Predictive entropy search for multi-objective bayesian optimization. In International conference on machine learning (pp. 1492-1501). PMLR.
>
> [7] Takeno, S., Tamura, T., Shitara, K., & Karasuyama, M. (2022, June). Sequential and parallel constrained max-value entropy search via information lower bound. In International Conference on Machine Learning (pp. 20960-20986). PMLR.
>
> [8] Suzuki, S., Takeno, S., Tamura, T., Shitara, K., & Karasuyama, M. (2020, November). Multi-objective Bayesian optimization using Pareto-frontier entropy. In International conference on machine learning (pp. 9279-9288). PMLR.
>
> [9] Tu, B., Gandy, A., Kantas, N., & Shafei, B. (2022). Joint entropy search for multi-objective Bayesian optimization. Advances in Neural Information Processing Systems, 35, 9922-9938.
>
> [10] Fernández-Sánchez, D., Garrido-Merchán, E. C., & Hernández-Lobato, D. (2020). Improved max-value entropy search for multi-objective bayesian optimization with constraints. arXiv preprint arXiv:2011.01150.
>
> [11] Garrido-Merchán, E. C., Fernández-Sánchez, D., & Hernández-Lobato, D. (2023). Parallel predictive entropy search for multi-objective Bayesian optimization with constraints applied to the tuning of machine learning algorithms. Expert Systems with Applications, 215, 119328.
>
> [12] Takeno, S., Fukuoka, H., Tsukada, Y., Koyama, T., Shiga, M., Takeuchi, I., & Karasuyama, M. (2020, November). Multi-fidelity Bayesian optimization with max-value entropy search and its parallelization. In International Conference on Machine Learning (pp. 9334-9345). PMLR.

---

### Official Review · Reviewer_WKw4 · 2025-10-31

**Soundness:** 3
**Presentation:** 2
**Contribution:** 2
**Rating:** 4
**Confidence:** 3

**Summary:**

The authors address the inefficiency of complex Monte Carlo approximations for Entropy Search (ES)-based acquisition functions in Bayesian Optimization (BO), which cause numerical errors and require cumbersome hand-crafted implementations. It proposes a two-stage amortization strategy using Prior-data Fitted Networks (PFNs): a base PFN is first trained to condition on information about the optima, and then the α-PFN is trained to predict the expected information gain using the information gains computed by the base PFN, enabling approximation in a single forward pass.

Key contributions:
1) It innovatively applies PFNs to amortize the approximation of ES variants (Predictive Entropy Search, Max-value Entropy Search, Joint Entropy Search), replacing costly sampling-based methods with a fast single forward pass and achieving speedups of at least 12 times (up to 41.2 times for 8-dimensional problems);
2) It supports fully Bayesian Gaussian Process models by integrating hyperparameter uncertainty, a capability rarely seen in existing ES implementations;
 3) Empirically, it matches the performance of state-of-the-art ES methods on synthetic Gaussian Process benchmarks and real-world hyperparameter optimization tasks from LC-Bench while maintaining high computational efficiency.

**Strengths:**

This paper proposes a two-stageα-PFN framework, which uses PFN amortized approximation to break through the limitation that traditional Entropy Search (ES)-based acquisition functions rely on complex Monte Carlo sampling. It is compatible with PES/MES/JES variants and supports fully Bayesian Gaussian Processes. Experiments on synthetic data and LC-Bench tasks verify that its performance is comparable to that of mainstream ES methods, with a speedup of at least 12× (reaching 41.2× for 8-dimensional tasks). The paper has a clear logical structure and explicit details, providing a new paradigm for the research on efficient acquisition functions in Bayesian Optimization and reducing the computational cost of practical applications.

**Weaknesses:**

1. Insufficient generalization in high-dimensional scenarios: The model is only trained up to 6 dimensions and validated on 8-dimensional tasks, failing to cover higher dimensions (e.g., 50+ dimensions), and the reasons for performance degradation in high dimensions are not analyzed. It is recommended to supplement synthetic experiments for 10-50 dimensions, analyze the feature utilization efficiency of PFN, and conduct corresponding optimizations.

2. Limited coverage of real-world scenarios: The validation is only conducted on LC-Bench hyperparameter optimization tasks, without involving complex real-world scenarios such as multi-objective optimization and noisy data. It is recommended to expand to multi-objective Bayesian Optimization (BO) tasks, validate on real noisy datasets, and supplement noise-resistant strategies.

3. Unoptimized PFN architecture: A fixed 6-layer Transformer architecture is adopted, with no exploration of the impact of different numbers of layers or attention heads, nor any attempts at model lightweighting. It is recommended to conduct architecture ablation experiments to select the optimal configuration and attempt model distillation.

4. Incomplete baseline comparisons: Only traditional ES variants and Log EI are used for comparison, with no inclusion of emerging methods (e.g., BORE). It is recommended to supplement such baselines to clarify the technical positioning of α-PFN.

**Questions:**

1. Supplement high-dimensional performance decomposition experiments: Count α-PFN’s time in "acquisition function evaluation", "data preprocessing," and "model inference" for 10-50D synthetic tasks, compare with traditional ES. Analyze high-dimensional feature importance via attention visualization; add mutual information-based feature selection if redundant features exist.

2. Optimize synthetic trajectory parameters: Use grid search (instead of random sampling) to find optimal ranges based on real-task (e.g., LC-Bench) historical queries. Introduce DTW distance to verify trajectory consistency, improving PFN training data authenticity and model generalization.

3. Explore hyperparameter sampling optimization: Test 5/10/20/50 sampling times’ impact on α-PFN. Fix 10 times if performance is comparable to more samplings. Try gradient-based methods (e.g., HMC) to enhance hyperparameter coverage and fully Bayesian model accuracy.

4. Improve out-of-distribution domain shift: Add domain adversarial training or augment GP samples like Levy function for tasks (e.g., Levy 4D). Supplement "domain shift degree-model performance" curves to clarify α-PFN’s adaptation boundary.

---

> ### Author Response · Authors · 2025-11-18
> **Clarification questions**
>
> Dear reviewer,
>
> Thank you for your feedback! To make the best of the rebuttal period, we have a number of clarification questions below. We would greatly appreciate it if you can provide more detail so that we may thoroughly address your concerns. Many thanks in advance for your time and consideration.
>
> **Weaknesses 1**. The reviewer writes: "It is recommended to supplement synthetic experiments for 10-50 dimensions, analyze the feature utilization efficiency of PFN, and conduct corresponding optimizations."
>
> Could you please clarify what you mean by "analyze the feature utilization of PFN" and "conduct corresponding optimizations"? We also wonder if you could highlight any specific high-dimensional synthetic functions or datasets, as the behavior of ES algorithms with GP-priors like the RBF kernel, to the best of our knowledge, have not been evaluated by even the most recent works on such high-dimensional datasets (for example, see Hvarfner et al., 2022).
>
> [1] Hvarfner, C., Hutter, F., & Nardi, L. (2022). Joint entropy search for maximally-informed Bayesian optimization. Advances in Neural Information Processing Systems, 35, 11494-11506.
>
> **Questions 1.** The reviewer writes: "Analyze high-dimensional feature importance via attention visualization; add mutual information-based feature selection if redundant features exist."
>
> We aren’t sure how to interpret this comment, but think it might be related to Weakness (1).  What is the motivation for high-dimensional feature importance, attention visualization, and utilizing mutual information -based feature selection to reduce redundant features? Does the reviewer have any specific concerns about the PFN architecture? Note that, visualizing and analyzing feature importances will be a non-trivial analysis; do you have any related works that carry this out in the context of ES that could help us understand what you intend us to carry out?
>
> **Questions 2**. The reviewer writes: "Use grid search (instead of random sampling) to find optimal ranges based on real-task (e.g., LC-Bench) historical queries. Introduce DTW distance to verify trajectory consistency, improving PFN training data authenticity and model generalization."
>
> While we agree that synthetic trajectories can be improved, we do not see how it can be achieved using DTW. We think measuring the quality of trajectories is non-trivial, and definitely cannot be done using DTW. Dynamic Time Warping (DTW) is intended for comparing time series of different lengths, where it is natural to speed up or slow down the signal - which is not applicable to our trajectories. We will greatly appreciate it if you can elaborate on this point further.
>
> **Question 3.** The reviewer writes: "Explore hyperparameter sampling optimization: Test 5/10/20/50 sampling times’ impact on α-PFN. Fix 10 times if performance is comparable to more samplings. Try gradient-based methods (e.g., HMC) to enhance hyperparameter coverage and fully Bayesian model accuracy."
>
> Could you please elaborate? Which hyperparameters are you referring to? We don't understand what "5/10/20/50 sampling times" means in this context. With prior-data fitted networks, we pre-train transformers via a user-specified data generating process (i.e., a prior), and the transformer itself amortizes fully bayesian inference [2], which can be done via a single forward pass.  Therefore, we don’t understand how HMC can be applied for posterior inference. Could you clarify how you intend us to apply HMC?
>
> [2] Transformers can do Bayesian inference. S. Muller et al. ICLR, 2022.
>
> **Question 4.** The reviewer writes: "Add domain adversarial training or augment GP samples like Levy function for tasks (e.g., Levy 4D). Supplement "domain shift degree-model performance" curves to clarify α-PFN’s adaptation boundary."
>
> Our goal is to develop a novel way of amortizing entropy-search BO techniques that is able to optimize real-world tasks using standard GP priors. We enforce similar priors for PFNs and GPs for fair comparison in the experiments. Augmenting GP samples with observations from test functions themselves (such as Levy 4D) would therefore give PFNs an unfair advantage in our comparison with GPs, we believe. Do you agree? Why or why not?
>
> Could you please explain what “"domain shift degree-model performance" curves” and expand on what you mean by the alpha-PFN’s adaptation boundary?

---

> > ### Author Response · Authors · 2025-12-03
> >
> > We thank the reviewer for the various suggestions. Hopefully, the reviewer finds our benchmarks and baseline methods well-motivated. Moreover, we have included a runtime table to clarify the computational advantages of our methods.
> >
> >
> > **Not sufficiently real-world scenarios.** We would like to clarify that LC-Bench tasks are _noisy_, real-world hyperparameter optimization problems: each function evaluation is a model training run with stochasticity, so our experiments already include noise. Thus, we firmly believe these benchmarks should suffice to demonstrate the effectiveness of our method.
> >
> > Regarding the multi-objective optimization request: Multi-objective is a non-trivial extension and therefore clearly out of scope for our work. This is illustrated by various papers[1, 2, 3, 4, 5, 6], which have as a single topic to extend existing ES strategies to the multi-objective setting.
> >
> >
> > **The PFN architecture is unoptimized.** We are already showing a significant speed-up, without any optimizations suggested by the reviewer. We believe this clearly shows a strength for our work: we did not even attempt yet to make our method more computationally efficient (simply just applying the PFN framework), yet we are already achieving considerable speed-up. This clearly illustrates that our work is an interesting first step, leading to potentially much larger gains in the future.
> >
> >
> > **Not sufficient baselines.**
> > We do include the strongest baseline: Log EI (a more numerically stable variant of EI) that has shown to be State-of-the-Art [8, 10] practically across the board in GP-based BO. We show that our Alpha-PFN is competitive with Log EI. Therefore, we do not see how the inclusion of other baselines would affect the conclusion.
> >
> > Furthermore, the central premise of our paper is improving upon existing entropy search methods, our work includes the required baselines to enable an apples-to-apples comparison. These baselines included are highly optimized ES implementations. We do not claim these are the best possible BO methods. In fact, some features of other BO methods (orthogonal to our contribution) are likely to give them an edge.
> >
> > **Need for higher-dimensional experiments.** First, we would like to stress that many relevant HPO problems are often low dimensional (e.g., HPO) and evaluations up to 8 dimensions are hardly uncommon in BO literature. 50D is “high dimensional BO”, which is an active research area, presenting many challenges. Most of these challenges are orthogonal to the main contribution of our work, e.g., the search procedure and prior would need to be adapted accordingly. Since training models to support 50-dimensional settings would be computationally very demanding, we have decided not to carry this out in the limited time of the rebuttal - we postpone this to future work. We anticipate that further scaling may be non-trivial, this primarily because of computational and practical issues - not due to some intrinsic limitation of our approach, e.g., the latest TabPFN model [9] supports up to 2000 dimensions.
> >
> > **Timings.** Please find more complete timings in Table 4, Appendix J. Table 1 is updated with new, correct runtimes.
> >
> > **Synthetic trajectory optimization.** We agree that the synthetic trajectories could likely be optimized further, but the search for an optimal trajectory design would be very challenging. However, without extensive tuning, we are able to achieve competitive performance with our Alpha-PFN. We will leave trajectory tuning to future work, as there are many different ideas to explore, and this could easily become computationally demanding.
> >
> > **Levy-like functions and domain shift.** Adding Levy-like functions to the training set would change our setting from 'GP priors only' to a hybrid prior that already encodes specific test functions. While this is definitely possible and should provide strong performance (on Levy-like functions), our goal was to study BO under Fully bayesian GP priors, and we prefer to keep the prior clean rather than tailoring it to individual benchmark functions. Similarly, domain-adversarial training and a thorough domain-shift study would be non-trivial and resource intensive. While interesting, we see them as natural follow-up work, rather than something we can realistically include in this first study.
> >
> > ______________
> >
> > We once again thank the reviewer for their thorough feedback and hope that we have alleviated their concerns through our rebuttal.

---

> ### Author Response · Authors · 2025-12-03
> **References**
>
> [1] Hernández-Lobato, D., Hernandez-Lobato, J., Shah, A., & Adams, R. (2016, June). Predictive entropy search for multi-objective bayesian optimization. In International conference on machine learning (pp. 1492-1501). PMLR.
>
> [2] Takeno, S., Tamura, T., Shitara, K., & Karasuyama, M. (2022, June). Sequential and parallel constrained max-value entropy search via information lower bound. In International Conference on Machine Learning (pp. 20960-20986). PMLR.
>
> [3] Suzuki, S., Takeno, S., Tamura, T., Shitara, K., & Karasuyama, M. (2020, November). Multi-objective Bayesian optimization using Pareto-frontier entropy. In International conference on machine learning (pp. 9279-9288). PMLR.
>
> [4] Tu, B., Gandy, A., Kantas, N., & Shafei, B. (2022). Joint entropy search for multi-objective Bayesian optimization. Advances in Neural Information Processing Systems, 35, 9922-9938.
>
> [5] Fernández-Sánchez, D., Garrido-Merchán, E. C., & Hernández-Lobato, D. (2020). Improved max-value entropy search for multi-objective bayesian optimization with constraints. arXiv preprint arXiv:2011.01150.
>
> [6] Garrido-Merchán, E. C., Fernández-Sánchez, D., & Hernández-Lobato, D. (2023). Parallel predictive entropy search for multi-objective Bayesian optimization with constraints applied to the tuning of machine learning algorithms. Expert Systems with Applications, 215, 119328.
>
> [7] Fröhlich, L., Klenske, E., Vinogradska, J., Daniel, C., & Zeilinger, M. (2020, June). Noisy-input entropy search for efficient robust Bayesian optimization. In International Conference on Artificial Intelligence and Statistics (pp. 2262-2272). PMLR.
>
> [8] Ament, S., Daulton, S., Eriksson, D., Balandat, M., & Bakshy, E. (2023). Unexpected improvements to expected improvement for bayesian optimization. Advances in Neural Information Processing Systems, 36, 20577-20612.
>
> [9] Grinsztajn, L., Flöge, K., Key, O., Birkel, F., Jund, P., Roof, B., ... & Hutter, F. (2025). TabPFN-2.5: Advancing the State of the Art in Tabular Foundation Models. arXiv preprint arXiv:2511.08667.
>
> [10] Hvarfner, C., Hellsten, E., Nardi, L. Vanilla Bayesian Optimization Performs Great in High Dimensions. ICML 2024.

---

### Official Review · Reviewer_Xqe8 · 2025-10-31

**Soundness:** 2
**Presentation:** 2
**Contribution:** 2
**Rating:** 6
**Confidence:** 3

**Summary:**

The paper proposes α-PFN, a two-stage amortization approach that learns information-theoretic acquisition functions (PES, MES, JES) for Bayesian optimization (BO) via Prior-data Fitted Networks (PFNs). A base PFN is first trained to approximate the posterior predictive distribution conditioned on oracle information about the maximizer ((x^*)) and/or max value ((f^*)). A second network, α-PFN, is then trained to predict the expected information gain in a single forward pass, replacing Monte-Carlo–based ES approximations at inference time. The method targets fast, learnable ES-style acquisitions with **12×–41×** speed-ups while remaining competitive in regret on synthetic functions and LC-Bench HPO tasks; it further includes a fully Bayesian variant by training on GPs with hyperpriors. The paper provides a derivation showing that α-PFN’s output distribution matches the ES objective in mean under an exact-base-PFN assumption. Code and checkpoints are released via an anonymized repository.

**Strengths:**

1. **Clear conceptual advance (amortized ES):** Recasts PES/MES/JES evaluation as a learned acquisition problem; α-PFN directly outputs the information gain, eliminating test-time sampling required by standard ES implementations. This is cleanly described and visualized (Figs. 1–2).
2. **Theoretical alignment to ES:** The training objective for α-PFN minimizes the KL between the true information-gain distribution and the model; the mean of α-PFN’s output equals the ES acquisition (Eq. (6)–(8), “Insight 1”), assuming the base PFN is accurate. This connects the method to ES formally.
3. **Runtime gains:** Reported **12×–41×** speed-ups across 2D/4D/8D while maintaining competitive regret; substantial for high-throughput BO.
4. **Handling of fully Bayesian GPs:** Training on GP hyperpriors makes fully Bayesian ES tractable in this learned framework (single acquisition rather than hyperparameter-averaged acquisitions).
5. **Domain-shift aware training:** Introduces a synthetic trace generator to mimic the global→local search transition in practical BO, addressing the distributional mismatch seen when training on uniformly sampled inputs.
6. **Empirical scope & transparency:** Evaluations on GP priors (2D/6D), synthetic test functions up to 8D, and LC-Bench HPO; results include 100-run averages and error bands; code + checkpoints are provided; compute budget documented.

**Weaknesses:**

1. **Key assumption left unquantified:** The formal link requires that the base PFN accurately approximates (p(y\mid x,D,I)), yet no error bounds or calibration diagnostics are provided for this approximation. This weakens the theoretical guarantee from a practical standpoint.
2. **Out-of-distribution degradation:** The method can underperform on OOD settings (e.g., Levy 4D; late iterations on Ackley 8D), highlighting prior-mismatch sensitivity and challenges generalizing beyond the training envelope.
3. **Training compute and engineering overhead:** Precomputing RFF GP samples and maximizers is expensive (~**40k CPU hours** per setting), and α-PFN must be retrained per prior, limiting plug-and-play applicability.
4. **Comparative fairness of timing:** Speed-ups compare GP-CPU vs PFN-GPU. A more stringent comparison would include GPU-accelerated GPs (e.g., GPyTorch/BoTorch on GPU) and report per-candidate acquisition latency, not only per-BO-loop runtime.

**Questions:**

1. **Calibration/accuracy:** How close is the base PFN’s conditioned PPD to GP posteriors? Please provide calibration plots and KL/JS to a GP baseline on small problems.
2. **Target fidelity:** For select instances where GP entropies are computable, how closely do PFN-based entropies match?

---

> ### Author Response · Authors · 2025-12-03
>
> Thank you for your review and your suggestions. We have added visualizations to demonstrate how the $\alpha$-PFN's entropy reduction compares to the ground truth, as well as the estimated entropy reduction of a GP. Hopefully, this convinces the reviewer that our amortization scheme yields accurate estimates for the acquisition function.
>
>
>
> **How close does the PFN computed acquisitions match the groundtruth ones? / Is the PFN well-calibrated?** We have conducted a similar study as in [5], which qualitatively compares GP and PFN-based approximations with the exact entropy reduction of conditioning on the optimizer. This is done using an expensive rejection sampling Monte Carlo technique.
>
> In Figure 6 (Appendix K), we show that the JES $\alpha$-PFN approximation is more similar to the ground truth both in magnitude and its ability to capture the general shape of the acquisition function, compared to a high-budget GP-based approximation. Note that this would not be possible without a faithful representation of the predictive posterior distribution, as the entropy estimation is a downstream quantity of the posterior.
>
> We have added the entropy estimation notebook to the supplementary material, too. We will move this plot in the main text for the final version.
>
> **The PFN suffers on Out-of-distribution data.** We agree that this is a weakness, but not substantially moreso than it would be for a fully Bayesian GP with the same hyperparameter prior. While neither Levy or Ackley look like plausible GP samples under a standard hyperprior, the fact that we train on a fairly standard fully Bayesian GP prior makes us inherit its strengths, as well as its weaknesses.
>
> With that said, the mismatch can be addressed by training on more diverse priors, and this has indeed been done, e.g. in [3]. As our work is acquisition function-oriented (as opposed to modeling-oriented). As the acquisition function work we conduct is perfectly complementary to the chosen hyperprior, the mismatch can be addressed the same way as in [3], for any prior that the user deems appropriate.
>
> **Pregenerating data.** We stress that, in practice, the procedure only needs to be repeated when applying it to some novel/custom prior, for which no α-PFN has been trained. In a sense, it is analogous to custom kernel design in GPs. While we already consider a fully Bayesian GP prior with dimension-dependent lengthscale hyper, future work plans to fully leverage the PFN’s flexibility to adopt even more expressive and general priors (like [3]), reducing the need for retraining. We agree that generating GP samples and their maximizers beforehand requires substantial compute. We believe this could be alleviated through methods like [1]. But that would come with additional approximation.
>
> **Timings.** Please find more complete timings below, and also in Table 4, Appendix J. Table 1 is updated with new, correct runtimes.
>
> | | | Model Fit | | | Acq. Opt. | | | PPD Opt. | | | Total | | |
> |---|---|---|---|---|---|---|---|---|---|---|---|---|---|
> | | | MES | JES | PES | MES | JES | PES | MES | JES | PES | MES | JES | PES |
> | **2D** | GP-CPU | 6.16 | 5.11 | 5.41 | 6.835 | 8.386 | 9.098 | 0.30 | 0.24 | 0.32 | **13.30** | **13.73** | **14.83** |
> | | GP-GPU | 6.04 | 6.05 | 5.98 | 0.433 | 0.551 | 1.173 | 0.22 | 0.37 | 0.22 | **6.69** | **6.97** | **7.38** |
> | | PFN-CPU | 0.0 | 0.0 | 0.0 | 2.35 | 3.20 | 2.10 | 3.26 | 3.71 | 2.92 | **5.61** | **6.91** | **5.02** |
> | | PFN-GPU | 0.0 | 0.0 | 0.0 | 0.64 | 0.68 | 0.78 | 0.78 | 0.74 | 0.77 | **1.41** | **1.43** | **1.54** |
> | **4D** | GP-CPU | 6.17 | 7.56 | 9.27 | 16.094 | 33.597 | 32.789 | 0.48 | 0.68 | 0.58 | **22.74** | **41.83** | **42.64** |
> | | GP-GPU | 6.84 | 7.10 | 7.10 | 0.903 | 1.067 | 1.769 | 0.38 | 0.42 | 0.41 | **8.12** | **8.59** | **9.28** |
> | | PFN-CPU | 0.0 | 0.0 | 0.0 | 3.41 | 6.37 | 4.32 | 4.90 | 7.77 | 4.98 | **8.31** | **14.13** | **9.30** |
> | | PFN-GPU | 0.0 | 0.0 | 0.0 | 1.47 | 1.26 | 1.36 | 1.10 | 1.04 | 1.33 | **2.57** | **2.30** | **2.68** |
> | **8D** | GP-CPU | 8.75 | 7.59 | 7.93 | 44.076 | 59.245 | 30.095 | 0.89 | 0.91 | 0.93 | **53.71** | **67.74** | **38.96** |
> | | GP-GPU | 8.17 | 8.03 | 8.29 | 1.567 | 1.386 | 1.974 | 1.00 | 0.88 | 0.87 | **10.73** | **10.30** | **11.14** |
> | | PFN-CPU | 0.0 | 0.0 | 0.0 | 8.90 | 7.55 | 9.13 | 8.80 | 8.24 | 9.02 | **17.70** | **15.79** | **18.15** |
> | | PFN-GPU | 0.0 | 0.0 | 0.0 | 1.30 | 1.33 | 1.19 | 0.90 | 1.02 | 0.97 | **2.20** | **2.35** | **2.15** |
>
>
> _____________________
>
> We once again thank the reviewer for their feedback, and hope that the added tables and figures have demonstrates the validity of our approach, and strengthened the reviewer's perception of our paper.

---

> ### Author Response · Authors · 2025-12-03
> **References**
>
> [1] Chang, P. E., Loka, N., Huang, D., Remes, U., Kaski, S., & Acerbi, L. (2025). Amortized probabilistic conditioning for optimization, simulation and inference. AISTATS 2025.
>
> [2] Müller, S., Hollmann, N., Arango, S. A., Grabocka, J., Hutter, F. (2022). Transformers Can Do Bayesian Inference. The Tenth International Conference on Learning Representations.
>
> [3] Müller, S., Feurer, M., Hollmann, N., & Hutter, F. (2023, July). Pfns4bo: In-context learning for bayesian optimization. In International Conference on Machine Learning (pp. 25444-25470). PMLR.
>
> [4] Adriaensen, S., Rakotoarison, H., Müller, S., & Hutter, F. (2023). Efficient bayesian learning curve extrapolation using prior-data fitted networks. Advances in Neural Information Processing Systems, 36, 19858-19886.
>
> [5] Hernández-Lobato, J. M., Hoffman, M. W., & Ghahramani, Z. (2014). Predictive entropy search for efficient global optimization of black-box functions. NeurIPS 2014

---

### Official Review · Reviewer_HNpe · 2025-10-31

**Soundness:** 3
**Presentation:** 3
**Contribution:** 3
**Rating:** 6
**Confidence:** 4

**Summary:**

This paper focuses on the problem that information-theoretic acquisition functions such as Entropy Search (ES) in Bayesian Optimization (BO) rely on complex Monte Carlo approximations, leading to slow computation and susceptibility to numerical errors. It aims to significantly improve the evaluation efficiency of acquisition functions and reduce the computational overhead of BO while ensuring optimization performance. A two-stage amortization strategy (α-PFN) based on Prior-data Fitted Networks (PFNs) is proposed: first train a base PFN to approximate the posterior predictive distribution, then train the α-PFN to directly predict the value of the acquisition function, replacing complex approximations with a single forward pass. Experiments verify that the α-PFN achieves performance comparable to mainstream methods and achieves a substantial speedup

**Strengths:**

1. It is highly meaningful that the paper adopts a two-stage amortization strategy (α-PFN) based on Prior-data Fitted Networks (PFNs) to address issues such as slow computational speed and susceptibility to numerical errors in traditional entropy search methods.
2. The paper elaborates on the proposed method in detail.
3. The paper has a reasonable organizational structure.
4. The paper conducts extensive experiments to verify the proposed method.
5. Code for reproducing the experiments is provided.

**Weaknesses:**

1. The method proposed in the paper is very similar to InfoNet, and such methods should be discussed in detail.

2. Figure 1 could be more visually appealing.

3. The paper seems to have used a hyperparameter search method, but relevant detailed data are not presented.

[1] Hu, Z., Kang, S., Zeng, Q., Huang, K., & Yang, Y. (2024). Infonet: neural estimation of mutual information without test-time optimization. arXiv preprint arXiv:2402.10158.

**Questions:**

no

---

> ### Author Response · Authors · 2025-12-03
>
> Thank you for your review, and thanks for acknowledging the strengths of our paper. We have clarified the connection between our approach and InfoNet, and added a table of the hyperparameters we used upon the reviewer's request.
>
> **Weakness 1: InfoNet.** We agree that our work is similar to InfoNet and we will certainly add it to our related work section. The key difference between our work and that of InfoNet is that we are amortizing the Conditional Mutual Information, which corresponds to the expectation over several entropy search acquisition functions. Meanwhile, InfoNet is designed for estimating Mutual Information between arbitrary variables. If we were to apply InfoNet to approximate the entropy search values; it would still require expensive sampling of multiple optima $x*$ and $f*$, conditioned on the observed data $D_{trn}$. The Alpha-PFN amortizes these expensive samples, offering further speed-up using a second step of amortization. So, while InfoNet is more a general concept, the specificity of our approach allows us to amortize a downstream quantity for even greater computational gains.
>
> Furthermore, in BO, we always train a base model to estimate the PPDs; which is why it is more natural to compute the acquisition values of ES using the entropy (differences) of the PPD.
>
> **Weakness 2: Figure of the pipeline.** Thanks for your feedback. We will improve the clarity and appearance of the figure in the final version.
>
> **Weakness 3: Hyperparameter tuning.** We did very little hyperparameter tuning. Specifically, we did not perform automated tuning or extensive parameter sweeps. The limited variations we tried throughout the design process did typically not have a strong impact on performance. In fact, we ended up reusing most hyperparameters from the PFNs4BO paper [1]; except for some adjustments to accommodate the improved PFN architecture from TabPFNv2  [2], reducing the embedding size. We chose our batch size as large as possible to maximize throughput in training. Please see the table below; you can see we reuse most hyperparameters directly. We increased the amount of bins, because we anticipated that modeling required a higher resolution in our BO setting, and because we observed that acquisition values can have low spread or close to zero; but we never performed an extensive ablation.
>
> | Hyperparameters | PFN4BO | Our work |
> |---|---|---|
> | Transformer layers | 12 | 8 |
> | Learning rate | 1e-4 | 1e-4 |
> | Embedding size | 512 | 128 (updated; there was a mistake in our appendix, which mentioned 512 - updated in our next draft) |
> | Attention heads | 4 | 4 |
> | Optimizer | Adam, cosine-annealing | AdamW, cosine annealing |
> | Number of hidden units | 1024 | 1024 |
> | Number of bins | 1000 | 5000 (mistake in our appendix, was 1000, will be updated) |
> | Linear warmup epochs | 10% | 25% |
>
>
> ______________
>
> We once thank the reviewer for their feedback, and hope that we have clarified their outstanding concerns regarding our paper.
>
>
> [1] Müller, S., Feurer, M., Hollmann, N., & Hutter, F. (2023, July). Pfns4bo: In-context learning for bayesian optimization. In International Conference on Machine Learning (pp. 25444-25470). PMLR.
>
> [2] Hollmann, N., Müller, S., Purucker, L., Krishnakumar, A., Körfer, M., Hoo, S. B., ... & Hutter, F. (2025). Accurate predictions on small data with a tabular foundation model. Nature, 637(8045), 319-326.
>
> [3] Adriaensen, S., Rakotoarison, H., Müller, S., & Hutter, F. (2023). Efficient bayesian learning curve extrapolation using prior-data fitted networks. Advances in Neural Information Processing Systems, 36, 19858-19886.

---

### Meta-Review · Area_Chair_FbFr · 2026-01-07

**Summary:**

Reviewer HNpe: It is highly meaningful that the paper adopts a two-stage amortization strategy (α-PFN) based on Prior-data Fitted Networks (PFNs) to address issues. The paper elaborates on the proposed method in detail. The paper conducts extensive experiments to verify the proposed method. However, the reviewer still has some concerns on the weaknesses about discussion with similar methods, lack more experimental details.

 Reviewer Xqe8: The paper has the strengths on the clear conceptual advance, theoretical alignment to ES, runtime gains, domain-shift aware training, empirical scope and transparency. However, the reviewer still has some concerns on the weaknesses about key assumption left unquantified, out-of-distribution degradation, training compute and engineering overhead, comparative fairness of timing.

 Reviewer WKw4: This paper proposes a two-stageα-PFN framework. It is compatible with PES/MES/JES variants and supports fully Bayesian Gaussian Processes. Experiments on synthetic data and LC-Bench tasks verify that its performance is comparable to that of mainstream ES methods. The paper has a clear logical structure and explicit details, providing a new paradigm for the research on efficient acquisition functions in Bayesian Optimization and reducing the computational cost of practical applications. However, the reviewer still has some concerns on the weaknesses about  insufficient generalization in high-dimensional scenarios, limited coverage of real-world scenarios, unoptimized PFN architecture, and incomplete baseline comparisons.

Reviewer Udjn: This paper is clearly well-written and interesting as a practical verification, leveraging the transformer for BO. The speed up can be a solid benefit, particularly when the objective function evaluation cost is moderate. However, the reviewer still has some concerns on the weaknesses about lack of more extensive experiments, lack of more comparisons with baselines,  no discussion and comparison of studies that leverage neural networks in other ways.

 Reviewer PgkG: The paper proposes an original two-stage PFN framework to efficiently approximate expensive information-theoretic ACQFs in BO. The idea of amortizing PES/MES/JES computation via neural networks is creative and addresses a key bottleneck in practical BO. Theoretical analysis is rigorous and clearly connects the two PFN stages, demonstrating that the-PFN can recover the expected information gain under mild assumptions. However, the reviewer still has some concerns on the weaknesses about lack of more explanations on method details, highly resource-intensive training, improper comparison presentations.

**Reviewer Concerns:**

After carefully evaluating the rebuttals, I think the reviews from the Reviewer Xqe8 were addressed from the response.
For the remaining reviewer concerns, they are all not fully addressed.

**Reviewer Scores:**

For the  Reviewer  Xqe8, I think the reviewer may increase the rating or keep the rating unchanged based on the response.
For the remaining Reviewers, I think the reviewer may keep the rating unchanged based on the response.

---

### Decision · Program_Chairs · 2026-01-26

Reject